# Temporal regulation of axonal repulsion by alternative splicing of a conserved microexon in mammalian *Robo1* and *Robo2*

Verity Johnson[1†], Harald J Junge[1‡], Zhe Chen[1,2]*

[1]Department of Molecular, Cellular and Developmental Biology, University of Colorado, Boulder, United States; [2]Linda Crnic Institute for Down Syndrome, University of Colorado school of Medicine, Aurora, United States

*For correspondence:
zhe.chen@colorado.edu

Present address: †ArcherDx, Boulder, United States; ‡Department of Ophthalmology and Visual Neurosciences, University of Minnesota Medical School, Minneapolis, United States

Competing interests: The authors declare that no competing interests exist.

**Abstract** Proper connectivity of the nervous system requires temporal and spatial control of axon guidance signaling. As commissural axons navigate across the CNS midline, ROBO-mediated repulsion has traditionally been thought to be repressed before crossing, and then to become upregulated after crossing. The regulation of the ROBO receptors involves multiple mechanisms that control protein expression, trafficking, and activity. Here, we report that mammalian ROBO1 and ROBO2 are not uniformly inhibited precrossing and are instead subject to additional temporal control via alternative splicing at a conserved microexon. The NOVA splicing factors regulate the developmental expression of ROBO1 and ROBO2 variants with small sequence differences and distinct guidance activities. As a result, ROBO-mediated axonal repulsion is activated early in development to prevent premature crossing and becomes inhibited later to allow crossing. Postcrossing, the ROBO1 and ROBO2 isoforms are disinhibited to prevent midline reentry and to guide postcrossing commissural axons to distinct mediolateral positions.

DOI: https://doi.org/10.7554/eLife.46042.001

## Introduction

Temporal and spatial regulation of cell signaling ensures the fidelity of axon pathfinding, which is crucial for nervous system development and function. Multiple signaling pathways coordinate the projection of commissural axons across the CNS (central nervous system) midline, such that the axons are attracted as they approach and enter the midline, but subsequently become repelled in order to exit the midline and to never recross (*Evans and Bashaw, 2010*). Within the spinal commissural axons, Netrin/DCC (Deleted in colorectal carcinoma) signaling is activated to promote midline attraction while SLIT/ROBO (Roundabout) signaling that mediates repulsion is inhibited prior to crossing. Following crossing, SLIT/ROBO signaling becomes upregulated to facilitate midline expulsion and to block midline reentry (*Evans and Bashaw, 2010*).

SLIT/ROBO signaling is evolutionarily conserved and plays an important role in various biological processes, including cell migration, axon guidance, angiogenesis, and organogenesis (*Blockus and Chédotal, 2016*). Dysregulation of the SLIT/ROBO pathway has been implicated in several forms of cancer and neurological disorders (*Ballard and Hinck, 2012*; *Blockus and Chédotal, 2014*). In both *Drosophila* and vertebrates, control over ROBO repulsion is achieved through multiple mechanisms. In *Drosophila*, Comm blocks ROBO insertion into the axonal surface (*Keleman et al., 2002*; *Keleman et al., 2005*). In vertebrates, PRRG4 (Proline-rich and Gla domain 4) represses ROBO1 surface trafficking, whereas USP33 (Ubiquitin-specific peptidase 33), RabGDI/GDI1 (GDP dissociation inhibitor 1), and CLSTN1 (Calsyntenin1) stabilize or promote ROBO1 surface localization

(*Alther et al., 2016*; *Justice et al., 2017*; *Philipp et al., 2012*; *Yuasa-Kawada et al., 2009*). In addition, miR-92 has been shown to suppress vertebrate ROBO1 protein translation (*Yang et al., 2018*). Without affecting ROBO1 protein expression or trafficking, fly *Robo2* and mammalian *Robo3* have been shown to inhibit *Robo1* activity using different mechanisms (*Evans et al., 2015*; *Sabatier et al., 2004*). Mammalian *Robo3* has also been suggested to promote midline crossing by facilitating Netrin/DCC attraction (*Zelina et al., 2014*).

The mammalian *Robo* genes undergo alternative splicing to produce variants with complex expression patterns and guidance activities (*Chen et al., 2008*; *Clark et al., 2002*; *Dalkic et al., 2006*; *Yuan et al., 1999*). *Robo1* and *Robo2* share a homologous alternative exon 6b, which is 9 and 12 bp in length, respectively. Recent studies demonstrate that alternatively spliced 'microexons' ($\leq$51 bp) are highly conserved in the nervous system and are frequently misregulated in autistic individuals (*Irimia et al., 2014*; *Li et al., 2015*). Whether the alternative splicing of microexon 6b in *Robo1* and/or *Robo2* contributes to the dynamic regulation of midline repulsion was until now completely unknown.

Here, we report that the alternative splicing of *Robo1* and *Robo2* (referred to herein as *Robo1/2*) at microexon 6b is crucial for axon guidance and is controlled by the NOVA (Neuro-oncological ventral antigen) family of splicing factors, which are neural-specific KH (hnRNP K homology)-type RNA-binding proteins (*Darnell, 2006*). We show that loss of *Nova1 and Nova2* (referred to herein as *Nova1/2*) alters the expression of exon 6b and leads to severe midline crossing and postcrossing guidance defects. Genetically restoring the normal expression profiles of *Robo1/2* exon 6b is able to reverse these defects in *Nova1/2* mutants. Interestingly, exon 6b alternative isoforms display distinct guidance activities and their production is developmentally regulated. Consequently, ROBO-mediated repulsion is not uniformly repressed precrossing as previously believed, but is instead activated initially to block premature crossing and is sufficiently blocked during crossing. Together, our study demonstrates that mammalian ROBO1/2 are subject to complex regulation, which is coordinated by alternative splicing, protein translation and trafficking, and activity inhibition by *Robo3*.

## Results

### Double knockout of *Nova1/2* disrupts midline crossing and postcrossing guidance of spinal commissural axons

We reported previously that *Nova1/2* double knockout (dKO) dampens DCC signaling by reducing the full-length $Dcc_{long}$ isoform while increasing the truncated $Dcc_{short}$ isoform, which results in delayed commissural neuron migration and axonal projection toward the midline (*Leggere et al., 2016*). As the defect is partial and is somewhat alleviated over time (from E10.5 to E12.5), some *Nova1/2* dKO axons are able to reach the midline at later stages (*Leggere et al., 2016*). We thus further examined midline crossing and postcrossing trajectories of axons in *Nova1/2* dKO embryos using DiI labeling. As subpopulations of axons follow different postcrossing trajectories (*Kadison and Kaprielian, 2004*), we selected medially located spinal neurons for comparison. At E12.5, we found that approximately half of the axons that had reached the midline failed to cross and turned longitudinally on the ipsilateral side in *Nova1/2* dKO embryos (*Figure 1*). In addition, some postcrossing axons projected away from the midline at greater angles in *Nova1/2* dKO mutants than in the wildtype (WT) controls (*Figure 1*). Thus, loss of *Nova1/2* causes additional guidance abnormalities in spinal commissural axons in addition to those we reported previously.

To determine if failed crossing in *Nova1/2* dKO mutants also results from reduced DCC attraction, we constructed *Dcc; Nova1; Nova2* triple KO mutants. If loss of *Dcc* is fully responsible for the blocked crossing, then the triple KO should cause no more severe defect than *Dcc* single KO. Using anti-ROBO3 to label axons approaching and crossing the midline, we found that *Nova1/2* dKO and *Dcc* single KO embryos exhibited reduced ventral commissures as previously reported (*Leggere et al., 2016*; *Xu et al., 2014*), and that *Dcc; Nova1; Nova2* triple KO embryos had even thinner commissures than *Dcc* single KO mutants (*Figure 2A,B*). This suggests that loss of *Nova1/2* is likely to affect additional guidance signaling besides reducing DCC attraction.

Blocked midline crossing has also been reported in *Robo3* KO embryos, where all commissural axons fail to cross and become abnormally sensitive to the SLIT2 repellent in explants (*Sabatier et al., 2004*). The crossing defect in *Robo3* KO mutants can be partially rescued by

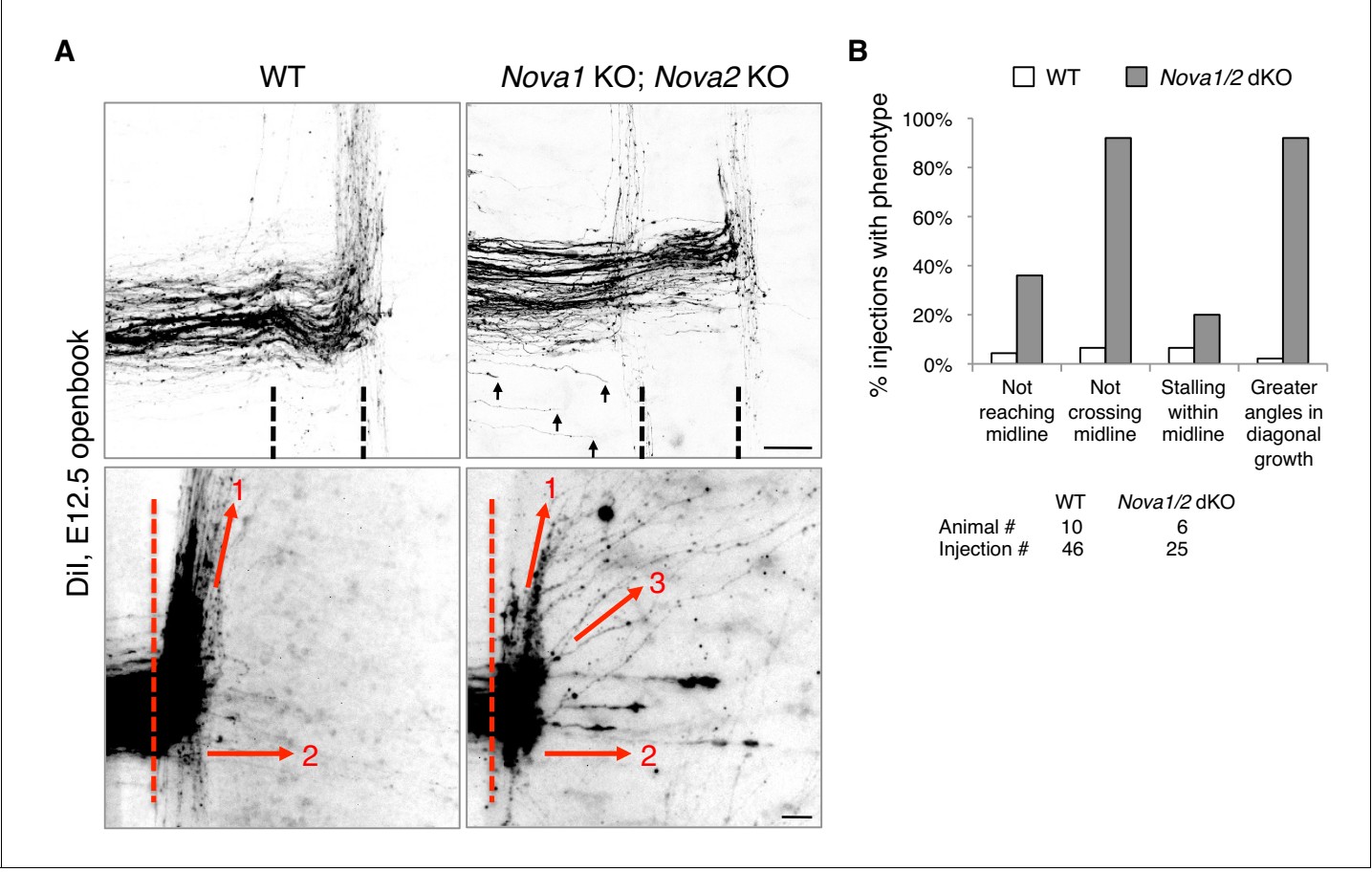

**Figure 1.** *Nova1/2* double knockout (dKO) disrupts commissural axon midline crossing and postcrossing guidance. (**A**) DiI labeling of E12.5 mouse spinal cords in openbook preparations. The top panel shows confocal micrographs of axons at the midline. *Nova1/2* dKO partially blocked midline crossing. Some axons had not yet reached the midline in *Nova1/2* dKO embryos (arrows), consistent with a reduction in DCC-mediated attraction as previously reported (*Leggere et al., 2016*). The bottom panel shows postcrossing axons that projected away from the midline. Some axons projected diagonally (arrow 1) and some projected straight away (arrow 2). In *Nova1/2* dKO embryos, some axons projected diagonally at greater angles from the midline than normal (arrow 3). Dashed lines indicate midline boundaries. Scale bars, 50 µm. (**B**) Quantification of the guidance defects from DiI tracing. Data are shown as the percentage of injection sites with the stated phenotypes. Midline recrossing was not observed in WT or *Nova1/2* dKO embryos. The trajectories of postcrossing axons along the rostrocaudal axis were comparable between WT and *Nova1/2* dKO embryos.
DOI: https://doi.org/10.7554/eLife.46042.002

The following source data is available for figure 1:

**Source data 1.** DiI tracing.
DOI: https://doi.org/10.7554/eLife.46042.003

deleting *Robo1/2* together, which allows about half of the axons to cross (*Jaworski et al., 2010*). Deleting *Robo1* alone has a mild rescue effect and deleting *Robo2* alone cannot rescue (*Jaworski et al., 2010*; *Sabatier et al., 2004*). These findings suggest that in the absence of *Robo3*, elevated ROBO1/2 activity can block midline crossing. To determine if elevated ROBO repulsion also blocks crossing in *Nova1/2* dKO embryos, we constructed *Nova1; Nova2; Robo1* triple KO mutants. *Robo1* single KO embryos have a normal commissure size (*Jaworski et al., 2010*), and by anti-ROBO3 staining, we found that *Nova1; Nova2; Robo1* triple KO mutants had slightly thicker commissures than *Nova1/2* dKO embryos (*Figure 2A,B*). Using DiI labeling, we further examined midline crossing in *Nova1; Nova2; Robo1* triple KO embryos and found that most of the axons that were able to reach the midline were also able to cross (*Figure 2C,D*). Thus, the blocked midline crossing in *Nova1/2* dKO embryos is most likely caused by elevated ROBO repulsion.

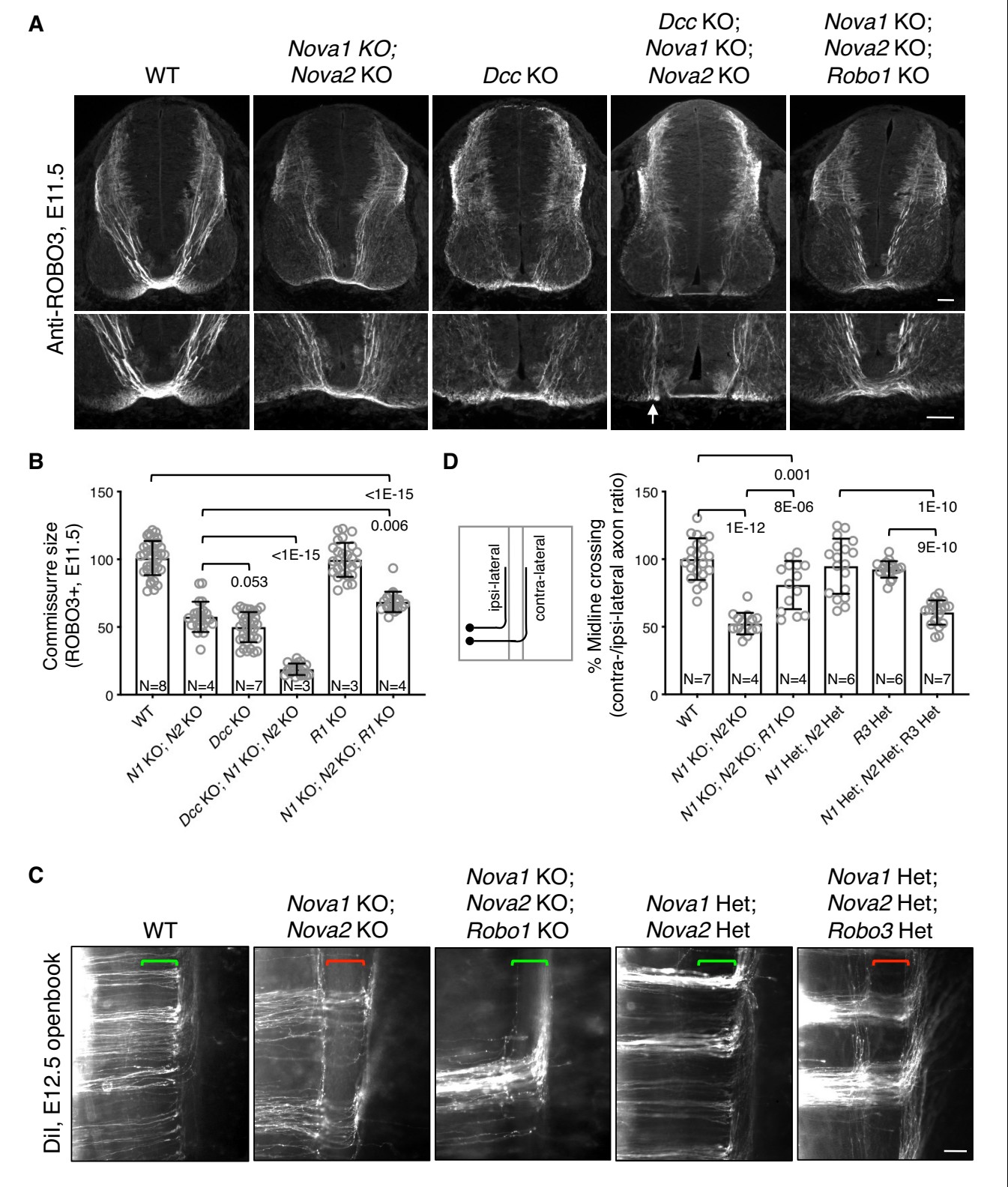

**Figure 2.** Midline crossing defect in *Nova1/2* dKO embryos results from elevated ROBO repulsion. (**A**) Anti-ROBO3 staining of commissural axons in transverse sections of E11.5 spinal cords. The bottom panel shows closeup images of the floorplate. *Nova1/2* dKO and *Dcc* KO embryos had reduced ventral commissures. *Dcc; Nova1; Nova2* triple KO mutants had even thinner commissures than *Dcc* KO embryos, and some axons appeared to remain on the ipsilateral side without entering the midline (arrow) in the triple KO mutants. In contrast, *Nova1; Nova2; Robo1* triple KO mutants had a slightly

*Figure 2 continued on next page*

*Figure 2 continued*

increased ventral commissure size compared to *Nova1/2* dKO embryos. Scale bars, 50 μm. (B) Quantification of the ventral commissure size in A. Data are normalized to the WT and are represented as the mean ± SD (one-way ANOVA and Bonferroni post test; animal numbers and p values are indicated). (C) DiI labeling of E12.5 mouse spinal cords in openbook preparations. In *Nova1/2* dKO embryos, about half of the axons arriving at the midline did not project across. This defect was alleviated by *Robo1* KO. *Nova1/2* dHet and *Robo3* Het, which were phenotypically normal on their own, synergistically blocked midline crossing. Brackets indicate the midline. Scale bar, 50 μm. (D) Quantification of midline crossing in C. Data are normalized to the WT and are represented as the mean ± SD (one-way ANOVA and Bonferroni post test; animal numbers and p values are indicated).

DOI: https://doi.org/10.7554/eLife.46042.004

The following source data is available for figure 2:

**Source data 1.** Commissure size and midline crossing.
DOI: https://doi.org/10.7554/eLife.46042.005

To understand the genetic relationship between *Nova1/2* and *Robo3* in regulating midline crossing, we constructed triple mutants that are heterozygous (Het) for deletions in all three genes. *Nova1/2* double Het (dHet) or *Robo3* single Het mutants did not exhibit any midline crossing defect (*Figure 2C,D*) (*Sabatier et al., 2004*). However, the triple Het mutants displayed midline crossing failure as severe as in *Nova1/2* dKO embryos (*Figure 2C,D*). Therefore, *Nova1/2* and *Robo3* have a synergistic interaction with regard to repressing midline repulsion and allowing axonal entry.

### *Nova1/2* double knockout disrupts *Robo1/2* alternative splicing

To determine how NOVA1/2 RNA-binding proteins may control *Robo* genes, we examined *Robo-1, -2, and -3* expression and alternative splicing in *Nova1/2* mutants. We previously reported that *Nova1/2* are both expressed by commissural neurons and their progenitors, and that *Robo3* expression and alternative splicing are not altered by *Nova1/2* dKO (*Leggere et al., 2016*). With respect to *Robo1* and *Robo2*, quantitative RT-PCR and western blotting analyses showed that the total mRNA and protein levels were not altered by *Nova1/2* dKO (*Figure 3—figure supplement 1A,B*). In addition, quantitative and semi-quantitative RT-PCR showed that most alternative sequences in the promoter and coding regions were not affected; however, the inclusion of microexon 6b in both *Robo1* and *Robo2* was significantly increased in *Nova1/2* dKO embryos (*Figure 3A,B*; *Figure 3—figure supplement 1C*). The inclusion of exon 6b was sensitive to *Nova1/2* gene copy number (*Figure 3—figure supplement 1D*), which is consistent with the dose-sensitive interaction between NOVAs and their targets (*Darnell, 2006*). Similar to the DCC-related guidance defect (*Leggere et al., 2016*), both *Nova1 Het; Nova2 KO* and *Nova1/2* dKO embryos had midline crossing failure, with the defect being more severe in *Nova1/2* dKO animals, whereas the other genotype combinations, including *Nova1/2* dHet mutants, were phenotypically normal. Hereafter, we refer to the *Robo1/2* isoforms that exclude and include exon 6b as e6b- and e6b+, respectively. Exon 6b encodes a short linker between the extracellular third and fourth Ig domains, which is conserved in mammals and chickens (*Figure 3C*). The inclusion of the alternative exon 21 in *Robo2*, which encodes an intracellular region between the CC1 and CC2 domains, was somewhat reduced in *Nova1/2* dKO embryos (*Figure 3A, B*).

Using ex vivo splicing assays, we tested if NOVAs directly regulate *Robo1/2* alternative splicing. We constructed splicing reporters for exon 6b (*Figure 3D*; *Figure 3—figure supplement 1E*; also see Materials and methods) and examined their alternative splicing in COS-1 cells. When an empty expression-control vector was coexpressed with the reporters, two RT-PCR products, corresponding to e6b- and e6b+, were detected. When *Nova1* or *Nova2* was overexpressed, e6b+ was reduced, consistent with the observation that e6b+ was abnormally increased in *Nova1/2* dKO mutants (*Figure 3A,B*). When candidate NOVA-binding sites in either intron 6 or intron 6b were mutated from YCAY (Y = C/U) to YAAY (*Buckanovich and Darnell, 1997*), NOVA-mediated repression of eb6+ was attenuated. Mutations in both introns almost completely blocked NOVAs from reducing e6b+ (*Figure 3D*). Thus, NOVAs bind to intron sequences flanking exon 6b and prevent its inclusion into the mRNA product. Using the same approach, we examined NOVA regulation of *Robo2* exon 21. We found that *Nova1* or *Nova2* overexpression did not alter the exon 21 splicing pattern, and that mutating candidate NOVA-binding sites in intron 20 also had no effect (*Figure 3— figure supplement 1F*). Thus, NOVAs by themselves do not appear to directly control exon 21

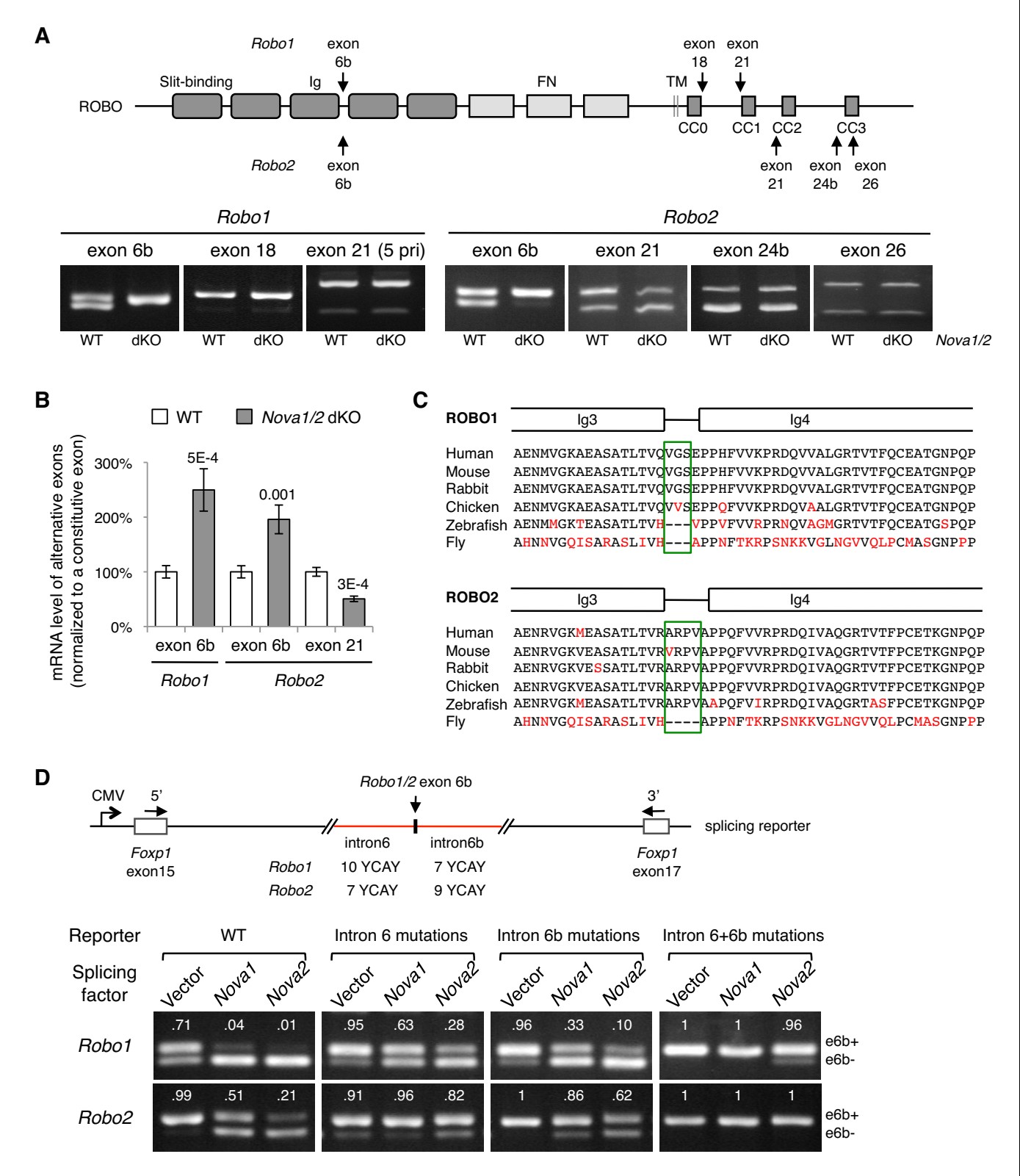

**Figure 3.** *Nova1/2* regulate *Robo1/2* alternative splicing. (**A**) Alternative splicing of *Robo1/2* exons assessed by semi-quantitative RT-PCR using E11.5 dorsal spinal cord (where commissural neurons are largely located). Schematic shows ROBO1/2 protein domains and the locations of alternatively spliced sequences. Ig, immunoglobulin domain; FN, fibronectin domain; TM, transmembrane domain; CC0-CC3, conserved cytoplasmic domain 0–3. The first Ig domain interacts with SLIT (**Morlot et al., 2007**). (**B**) Quantitative RT-PCR analyses of *Robo1/2* alternative exons whose expression was

*Figure 3 continued on next page*

*Figure 3 continued*

altered by *Nova1/2* dKO. Data are normalized to the WT and are represented as the mean ± SD (Student's t-test, two-tailed and unpaired; n = 3 animals; p values are indicated). (C) Alignment of peptide sequences encoded by *Robo1/2* exon 6b. Exon 6b encodes a short linker (boxed sequences) between the Ig3 and Ig4 domains. Identical residues are shown in black and non-identical ones in red. *Drosophila* ROBO1 was used for alignment, as it is homologous to both ROBO1 and ROBO2 in vertebrates. (D) Splicing assays of *Robo*1/2 exon 6b in COS-1 cells, where a splicing reporter was coexpressed with *Nova1* or *Nova2* and the levels of splice variants were assessed by RT-PCR. Flanking introns of exon 6b in the reporter (in red) contain candidate NOVA-binding sites (7–10 YCAY repeats; Y = C/U; mutations created YAAY repeats; see Materials and methods and *Figure 3—figure supplement 1E*). Alternatively spliced sequences were detected by semi-quantitative RT-PCR using 5' and 3' primers as indicated. Numbers in the electrophoresis images indicate the e6b+ level normalized to the total amount of both isoforms. *Nova1/2* normally repressed the e6b+ levels of both *Robo1* and *Robo2*. Mutating candidate NOVA-binding sites in either flanking intron partially blocked NOVA activity. Eliminating all NOVA-binding sites caused e6b+ to be exclusively expressed.

DOI: https://doi.org/10.7554/eLife.46042.006

The following figure supplement is available for figure 3:

**Figure supplement 1.** Alternative splicing of *Robo1/2*.

DOI: https://doi.org/10.7554/eLife.46042.007

alternative splicing. They may cooperate with or function indirectly through other splicing factors to exert their effect on exon 21.

## Restoring *Robo1/2* microexon 6b expression levels rescues *Nova1/2* dKO defects

To definitively determine if increased inclusion of exon 6b is responsible for the blocked midline crossing in *Nova1/2* dKO embryos, we genetically reduced exon 6b levels by deleting one copy of the exon from the *Robo1/2* genomic sequences using the CRISPR/Cas9 technology (*Figure 4A*; *Figure 4—figure supplement 1A*). In the *Nova1/2* dKO background, the WT allele predominantly expressed e6b+ (*Figure 3A*; *Figure 4A*), whereas the deletion allele only produced e6b- (*Figure 4—figure supplement 1D*). The combination between the WT and deletion alleles generated the two isoforms at a similar ratio to that in the WT at E11.5 (*Figure 4A*). Using this approach, we successfully restored the e6b+/e6b- levels of *Robo1* and *Robo2*, either alone or in combination, in *Nova1/2* dKO embryos. Deleting exon 6b in *Robo1* and/or *Robo2* did not alter the total mRNA or protein levels (*Figure 4—figure supplement 1B,C*), and did not affect the splicing of the surrounding areas or of other alternative exons (*Figure 4—figure supplement 1D*).

Using DiI labeling, we found that reducing *Robo1(e6b+)* alone partially restored midline crossing in *Nova1/2* dKO mutants, whereas reducing *Robo2(e6b+)* alone had no effect (*Figure 4B,C*). Reducing both *Robo1(e6b+)* and *Robo2(e6b+)* together further rescued the midline crossing defect (*Figure 4B,C*).

After crossing the midline, commissural axons join longitudinal tracts that traverse parallel to the floorplate at different dorso-ventral positions. SLIT/ROBO signaling has been shown to direct the lateral positioning of longitudinal axons (*Farmer et al., 2008*; *Jaworski et al., 2010*; *Kim et al., 2011*; *Long et al., 2004*). In *Nova1/2* dKO embryos, we found that some DiI-labeled postcrossing axons steered away from the midline at abnormally larger angles, raising the possibility of abnormal lateral positioning (*Figure 1*). As DiI traced only a subset of axons, we used the anti-L1 marker that labels all postcrossing commissural axons as well as ipsilateral-projecting axons to further assess the dorso-ventral positioning, as previously described (*Jaworski et al., 2010*). As there are fewer postcrossing axons in *Nova1/2* dKO embryos due to fewer axons reaching the midline, we compared the ratio between the ventral and lateral funiculi within the same section to determine the relative distribution of longitudinal axons. We found that the ventral tract close to the midline was reduced, whereas the lateral tract was enlarged in *Nova1/2* dKO spinal cords (*Figure 4B,C*), suggesting that some longitudinal axons projected further away from the midline. We also found that reducing *Robo1(e6b+)* alone in *Nova1/2* dKO embryos partially reversed this defect, whereas reducing *Robo2(e6b+)* alone had no effect (*Figure 4B,C*). Reducing both *Robo1(e6b+)* and *Robo2(e6b+)* largely rescued the defect (*Figure 4B,C*). The same degree of rescue was also observed in anti-ROBO1 labeled longitudinal axons (*Figure 4—figure supplement 2A,B*). Anti-ROBO2 staining was not used to assess the phenotype as ROBO2 is primarily expressed by axons in the lateral funiculi (*Figure 4—figure supplement 2C*). The overall patterns of ROBO1/2 expression, which are low on precrossing axons and

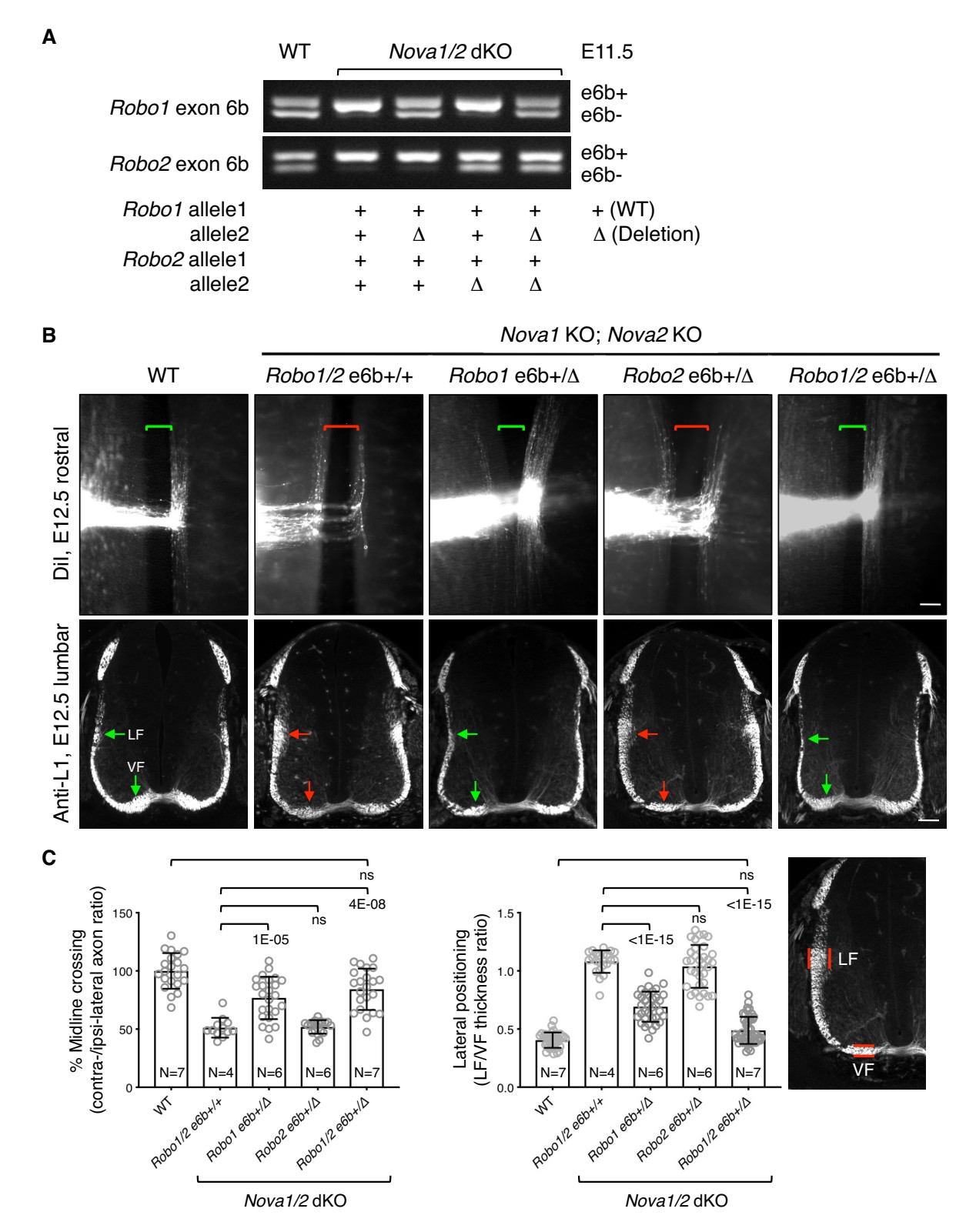

**Figure 4.** Restoring *Robo1/2* exon 6b levels rescues axon guidance defects in *Nova1/2* dKO embryos. (**A**) Alternative splicing of exon 6b in animals with or without the exon 6b deletion allele. Semi-quantitative RT-PCR was performed using RNA from E11.5 dorsal spinal cord. Deleting one copy of exon 6b from *Robo1/2* genomic DNA in *Nova1/2* dKO embryos restored e6b+ expression to a comparable level as in the WT. (**B**) DiI labeling of spinal cord openbooks (top panel) and anti-L1 staining of transverse spinal cord sections (bottom panel) in E12.5 animals with or without exon 6b deletion. As it is

*Figure 4 continued on next page*

*Figure 4 continued*

difficult to obtain a large number of compound mutants, the rostral half of the spinal cord was used for DiI labeling and the lumbar level was used for anti-L1 staining. Brackets indicate the midline. Arrows indicate the lateral funiculus (LF) and ventral funiculus (VF). Reducing *Robo1(e6b+)* partially rescued both midline crossing and lateral positioning defects in *Nova1/2* dKO embryos, while reducing *Robo2(e6b+)* alone did not rescue. Reducing *Robo1/2(e6b+)* together further rescued both defects. Scale bars, 50 µm. (**C**) Quantification of axon midline crossing and lateral positioning in B. As *Nova* deficiency reduces the number of postcrossing axons due to fewer axons reaching the midline, we compared the ratio between the thickness of the lateral and ventral funiculi within the same section. Data are represented as the mean ± SD (one-way ANOVA and Bonferroni post test; animal numbers and p values are indicated; ns, not significant).

DOI: https://doi.org/10.7554/eLife.46042.008

The following source data and figure supplements are available for figure 4:

**Source data 1.** Phenotypic rescues by exon 6b deletion.
DOI: https://doi.org/10.7554/eLife.46042.012
**Figure supplement 1.** Generation of *Robo1/2* exon 6b deletion alleles by CRISPR/Cas9.
DOI: https://doi.org/10.7554/eLife.46042.009
**Figure supplement 2.** Lateral positioning of ROBO1-expressing longitudinal axons in *Nova1/2* dKO embryos.
DOI: https://doi.org/10.7554/eLife.46042.010
**Figure supplement 3.** *Nova1/2* and *Dcc* mutants have distinct postcrossing guidance defects.
DOI: https://doi.org/10.7554/eLife.46042.011

are highly upregulated on postcrossing axons, were comparable between WT and *Nova1/2* dKO embryos (*Figure 4—figure supplement 2A,C*).

Netrin/DCC signaling has been shown to control postcrossing commissural axon trajectories in the hindbrain, but not in the spinal cord. In *Netrin1* or *Dcc* KO mutants, hindbrain commissural axons project at larger angles away from the midline, whereas spinal commissural axons have normal post-crossing projection (*Shoja-Taheri et al., 2015*). To investigate whether the lateral positioning defect in *Nova1/2* mutants can result from impaired DCC signaling, we examined anti-L1 staining in *Dcc; Nova1; Nova2* triple KO spinal cords. *Dcc* KO mutants had a slight increase in the lateral funiculi, which was much less severe than in *Nova1/2* dKO embryos (*Figure 4—figure supplement 3*). In *Dcc* KO embryos, some lateral axons evaded the medial spinal cord, which could result from axon defasi-culation and/or from a lack of restrictive signal from the medial spinal cord. Defasiculation is observed in precrossing axons in *Dcc* KO, but not in *Nova1/2* dKO embryos (*Leggere et al., 2016*; *Xu et al., 2014*), possibly due to the fact that DCC_short is still present in *Nova1/2* dKO mutants, which may be sufficient for axonal adhesion. In *Dcc; Nova1; Nova2* triple KO embryos, the lateral funiculi were enlarged and most lateral axons projected into the medial spinal cord. Thus, the defects in the triple KO mutants appear to be a combination of those seen in *Dcc* and *Nova1/2* KO mutants. Taken together, the lateral positioning defect in *Nova1/2* dKO embryos cannot result solely from a deficiency in DCC signaling, although loss of *Dcc* may contribute to a small degree.

## Microexon 6b splice variants of ROBO1/2 have distinct guidance activities

To distinguish between the in vivo activities of e6b- and e6b+, we compared the effects of ectopi-cally expressing the isoforms in chicken embryos. We found that *Robo1(e6b-)* expression partially blocked midline crossing, with ~30% of the axons failing to cross. *Robo1(e6b+)* further blocked mid-line crossing, with ~49% of the axons being unable to cross (*Figure 5A,B*). Thus, e6b+ appears to have a stronger repulsive effect than e6b-. We also overexpressed the *Robo1* isoforms in cultured mouse embryos. In *Robo3* Het embryos, which were used to lower the inhibition of exogenously expressed mouse *Robo1* cDNA and thus to potentiate the effect, we observed significant midline crossing defects (*Figure 5—figure supplement 1A*). *Robo1(e6b+)* overexpression blocked about half of the axons from crossing the midline, whereas *Robo1(e6b-)* overexpression caused a milder defect (*Figure 5—figure supplement 1A*), consistent with their effects in chickens.

In addition, *Robo1(e6b-)* ectopic expression in chicken embryos directed commissural axons that had reached the contralateral side or abnormally stayed on the ipsilateral side to travel closely to the midline within the ventral funiculi (*Figure 5A,B*). In contrast, *Robo1(e6b+)* ectopic expression guided axons further away from the midline to a more dorsolateral position than normal (*Figure 5A, B*). Thus, *Robo1(e6b-)* and *Robo1(e6b+)* have distinct activities in guiding postcrossing commissural

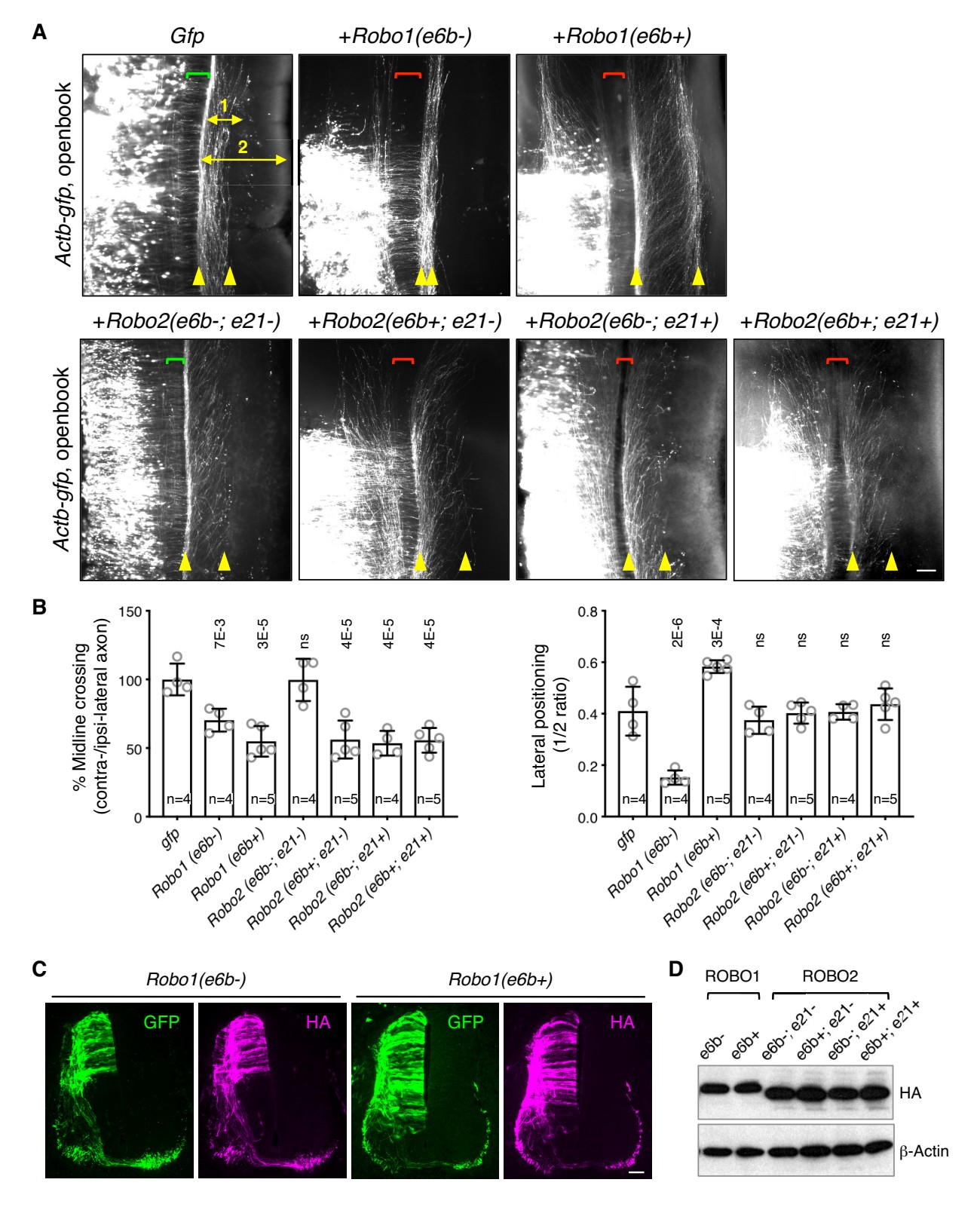

**Figure 5.** ROBO1/2 splice variants have distinct guidance activities. (**A**) Openbook preparations of chicken spinal cords electroporated with *Actb-gfp* or with different *Robo1* or *Robo2* isoforms. Transfected neuronal cell bodies are oriented to the left, and the rostral end of the spinal cord is pointing up. Brackets indicate the midline. Arrowheads indicate the medial and lateral positions of postcrossing axons. Distance one is between the midline and the dorsolateral-most longitudinal axons and distance two is the total height of the spinal cord. Scale bar, 50 µm. (**B**) Quantification of midline crossing and

*Figure 5 continued on next page*

*Figure 5 continued*

lateral positioning of commissural axons in A. Lateral positioning is quantified as the ratio between distances one and two. Data are represented as the mean ± SD (one-way ANOVA and Bonferroni post test; animal numbers and p values are indicated; ns, not significant). (**C**) Immunohistochemistry of transverse sections of chicken spinal cords electroporated with HA-tagged *Robo1* isoforms. Scale bar, 50 µm. (**D**) Western blotting analysis of lysates from chick spinal cords electroporated with HA-tagged *Robo1/2* isoforms.

DOI: https://doi.org/10.7554/eLife.46042.013

The following source data and figure supplement are available for figure 5:

**Source data 1.** Overexpression of *Robo1/2* isoforms.
DOI: https://doi.org/10.7554/eLife.46042.015
**Figure supplement 1.** ROBO1/2 splice variants have distinct guidance activities.
DOI: https://doi.org/10.7554/eLife.46042.014

axons. Using anti-L1 staining, we also examined the positioning of longitudinal axons in homozygous exon 6b deletion mice, and found that *Robo1* e6bΔ/Δ embryos, which produced exclusively e6b- (*Figure 4—figure supplement 1D*), had enlarged ventral funiculi (*Figure 5—figure supplement 1B*), consistent with the observation that e6b- directed axons close to the midline in chickens (*Figure 5A,B*).

As both exons 6b and 21 of *Robo2* were affected by *Nova1/2* dKO, we cloned cDNAs with all four combinations between the exons and tested their activities in chicken embryos. We found that ectopic expression of *Robo2(e6b-; e21-)* did not affect midline crossing (*Figure 5A,B*). In contrast, expression of *Robo2(e6b+; e21-)*, *Robo2(e6b-; e21+)*, or *Robo2(e6b+; e21+)* similarly blocked about half of the axons from crossing (*Figure 5A,B*). Thus, the inclusion of four amino acids in the extracellular domain from exon 6b or the inclusion of 42 amino acids in the intracellular domain from exon 21 similarly increases ROBO2 repulsion. The effects of these exon inclusions are neither synergistic nor additive. Taken together, in the absence of the exon 21 coding sequence, e6b+ appears to be more repulsive than e6b-. In the presence of exon 21, e6b- and e6b+ are comparable in their activities.

Ectopic expression of all *Robo2* isoforms directed postcrossing axons to comparable lateral positions as in GFP-expressing controls (*Figure 5A,B*), indicating similar activities among ROBO2 variants in guiding longitudinal axons. Consistent with these findings, *Robo2* e6bΔ/Δ homozygous mutants had normal dorso-ventral positioning of longitudinal tracts (*Figure 5—figure supplement 1B*).

Using immunohistochemistry and western blotting, we found that the electroporated *Robo1/2* isoforms were expressed at comparable levels in chicken embryos and that the exogenously expressed receptors could be detected on both pre- and post-crossing axons (*Figure 5C,D*).

## Microexon 6b splice variants of ROBO1/2 have distinct signaling properties

Using cell surface biotinylation assays, we found that the ROBO1 or ROBO2 isoforms were inserted into the surface of COS-1 cells at comparable levels (*Figure 6—figure supplement 1A*). We then compared the binding between the ROBO variants with an AP-SLIT2 (N-term) fusion protein in COS-1 cells. We found that the ROBO1 or ROBO2 isoforms had comparable binding affinities toward SLIT2 (N-term) (*Figure 6—figure supplement 1B*), consistent with the fact that the isoforms share the same SLIT-binding domain (*Figure 3A*) (*Morlot et al., 2007*).

The RHO family of small GTPases have been shown to mediate SLIT/ROBO signaling during cell migration and axon guidance (*Bashaw and Klein, 2010*). Activation of the SLIT/ROBO pathway inhibits CDC42, activates RAC1, and has distinct effects on RHOA in different cell types (*Fan et al., 2003*; *Guan et al., 2007*; *Wong et al., 2001*; *Yang and Bashaw, 2006*). Using small GTPase activation assays, we compared the downstream signaling of the ROBO1/2 isoforms. We found that both ROBO1 isoforms inhibited CDC42 and activated RAC1, with e6b+ exerting a stronger effect than e6b- (*Figure 6*). Among the four ROBO2 isoforms, e6b-; e21- had no effect on either CDC42 or RAC1, while the other three similarly inhibited CDC42 and activated RAC1 (*Figure 6*). We did not observe significant changes in RHOA activation in the COS-1 cells expressing any of the isoforms upon SLIT2 N-term stimulation. Taken together, *Robo2(e6b-;e21-)*, the only isoform that does not block midline crossing when ectopically expressed (*Figure 5*), has no effect on CDC42/RAC1

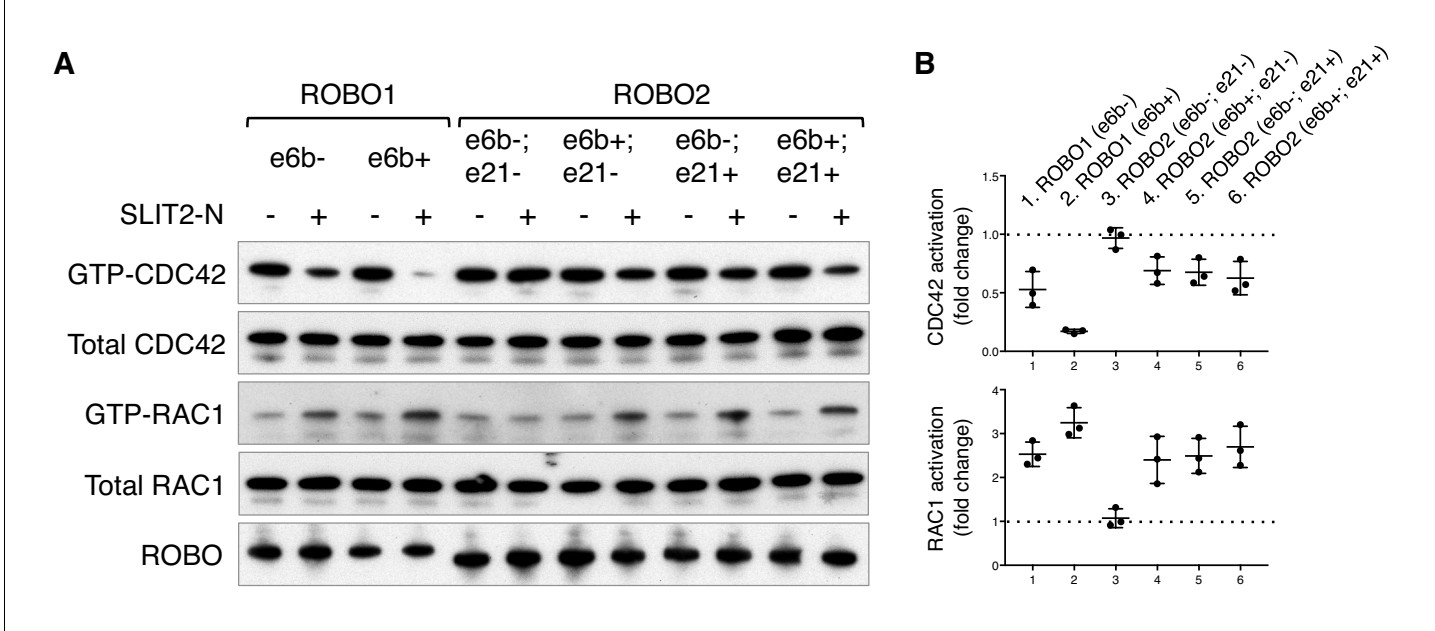

**Figure 6.** ROBO1/2 splice variants have distinct signaling activities. (**A**) Regulation of CDC42/RAC1 activation by ROBO1/2 isoforms in COS-1 cells. SLIT2 N-term was added at 500 ng/ml for 10 min at 37˚C. GTP-bound/activated CDC42 or RAC1 was pulled down by PAK-GST beads and detected by western blotting. (**B**) Data from three independent assays showing fold changes in CDC42/RAC1 activation upon SLIT2 N-term stimulation, represented as the mean ± SD.

DOI: https://doi.org/10.7554/eLife.46042.016

The following source data and figure supplement are available for figure 6:

**Source data 1.** Activation of small GTPases by ROBO1/2 isoforms.

DOI: https://doi.org/10.7554/eLife.46042.018

**Figure supplement 1.** ROBO1/2 splice variants bind to SLIT2 with similar affinities.

DOI: https://doi.org/10.7554/eLife.46042.017

(*Figure 6*). By contrast, other ROBO1/2 isoforms inhibit CDC42 and activate RAC1 following SLIT2 binding. Between the ROBO1 isoforms, e6b+ has a stronger effect than e6b-.

## Microexon 6b alternative splicing is developmentally controlled and is important for proper timing of midline crossing

Using rat embryos to enable separation of the dorsal and ventral halves of the spinal cord, we examined *Robo1/2* exon 6b alternative splicing and *Nova1/2* expression in commissural neurons (predominantly in the dorsal half) over developmental stages. In rats, neurogenesis in the dorsal spinal cord starts around E11, and midline crossing is almost complete by E14 (the equivalent stages are E9.5 to E12.5 in mice). We found that *Robo1/2(e6b+)* was the predominant form at E11, and that over time, *Robo1/2(e6b+)* decreased while *Robo1/2(e6b-)* increased (*Figure 7A*). The switch in the e6b isoform levels coincided with a steady increase in *Nova1/2* expression over this period (*Figure 7B*), which represses e6b+ production (*Figure 3D*).

Since the predominant expression of *Robo1/2(e6b+)* in *Nova1/2* dKO embryos blocked midline crossing despite the presence of other inhibitory factors of ROBO1/2 (*Figure 4*), we wondered if the expression of e6b+ during early development would help prevent premature midline crossing by commissural axons. Using anti-ROBO3 staining, we followed early projection of commissural axons at the lumbar level of E10.5 spinal cord, where and when many commissural axons had not yet reached the floor plate and appeared to be unfasciculated (*Figure 7C*). The size of the ventral commissure was small (*Figure 7D*). By contrast, in *Robo1* e6bΔ/Δ mutants lacking e6b+, the ventral commissure was much thicker and commissural axons were mostly fasciculated (*Figure 7C,D*). This suggests that a larger number of axons had reached and crossed the midline in the mutants than in the WT controls. We also found that *Robo2* e6bΔ/Δ mutants were indistinguishable from the WT,

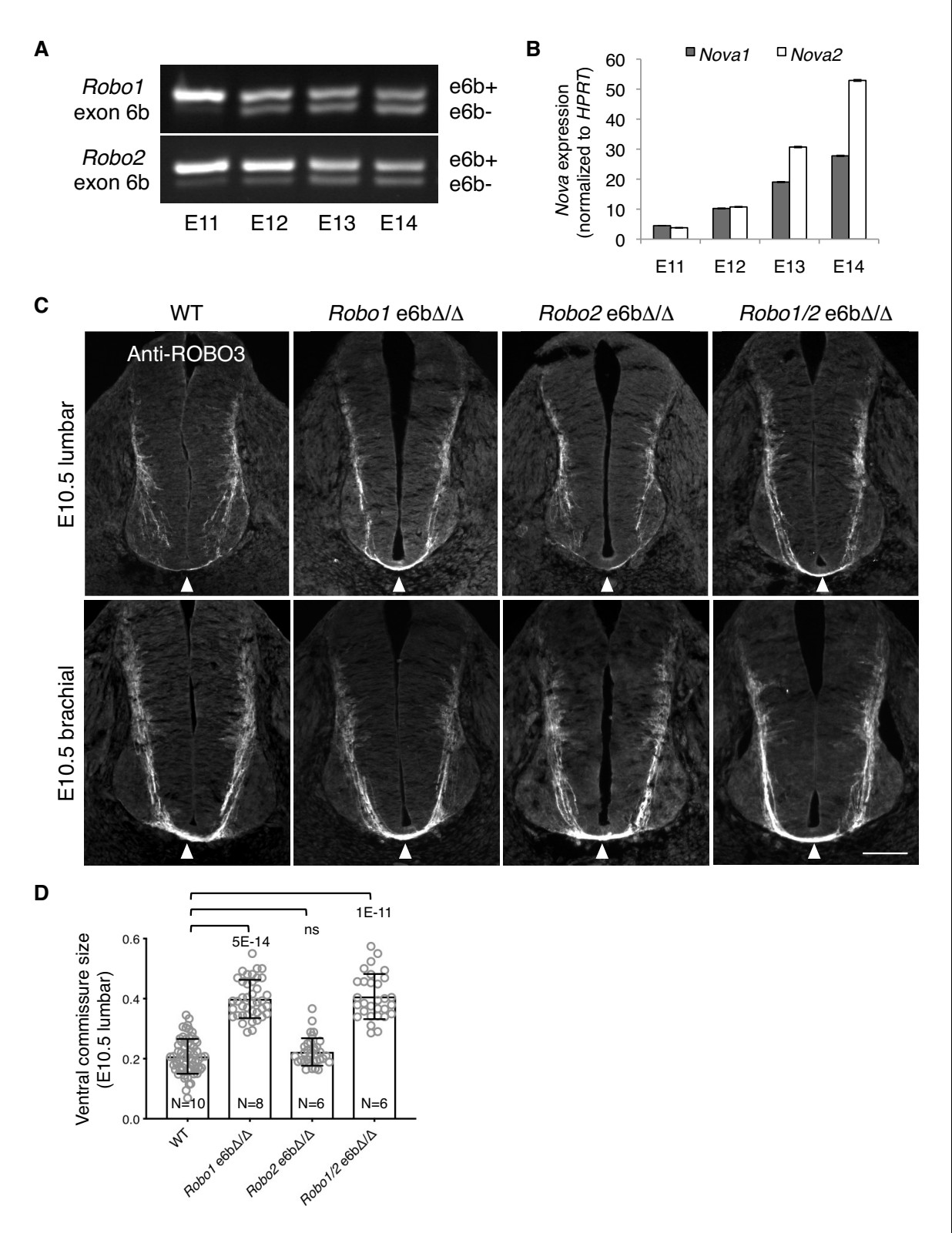

**Figure 7.** Alternative splicing of *Robo1/2* is developmentally regulated to ensure proper timing of midline crossing. (**A,B**) The expression levels of *Robo1/2* exon 6b and *Nova1/2*, respectively, in dorsal spinal cord from E11 to E14 rats (equivalent of E9.5 to E12.5 in mice), as measured by RT-PCR. Rat embryos were used to allow separation of the dorsal and ventral halves of the spinal cord. (**C**) Anti-ROBO3 staining of transverse sections of WT and *Robo* e6bΔ/Δ spinal cords at E10.5 lumbar (top) and brachial (bottom) levels. Arrowheads indicate the ventral commissure consisting of midline-
*Figure 7 continued on next page*

*Figure 7 continued*

crossing axons. The ventral commissures in *Robo1* e6bΔ/Δ and *Robo1/2* e6bΔ/Δ mutants were thicker than in the WT controls at the lumbar level. All genotypes had comparable ventral commissure thickness at the brachial level. Scale bar, 50 µm. (**D**) Quantification of the ventral commissure size in C. Data are represented as the mean ± SD (one-way ANOVA and Bonferroni post test; animal numbers and p values are indicated; ns, not significant).
DOI: https://doi.org/10.7554/eLife.46042.019
The following source data and figure supplement are available for figure 7:

**Source data 1.** Early projection of commissural axons in exon 6b deletion mutants.
DOI: https://doi.org/10.7554/eLife.46042.021
**Figure supplement 1.** Developmental expression of TAG1 in mouse embryos.
DOI: https://doi.org/10.7554/eLife.46042.020

whereas *Robo1/2* e6bΔ/Δ double mutants displayed a similar phenotype to that in *Robo1* e6bΔ/Δ single mutants (*Figure 7C,D*). Thus, *Robo1(e6b+)*, but not *Robo2(e6b+)*, appears to prevent premature midline crossing by commissural axons. To ensure that the mutant and WT embryos were comparable in their overall development, we examined embryos from multiple litters and selected embryos of similar sizes. As TAG1 expression by motor, commissural, and dorsal root ganglion (DRG) neurons is developmentally controlled and transient (*Dodd et al., 1988*) (*Figure 7—figure supplement 1A*), we also used anti-TAG1 staining to ensure that the WT and mutant spinal cords were of a comparable developmental stage (*Figure 7—figure supplement 1B*).

At the brachial level of E10.5 spinal cord, where neuronal development proceeds further than at the lumbar level, additional commissural axons were found to have crossed the midline based on anti-ROBO3 staining (*Figure 7C*). All exon 6b deletion mutants became indistinguishable from the WT controls with regard to the size of the ventral commissure (*Figure 7C*). Thus, the lack of *Robo1(e6b+)* allows commissural axons to cross the midline earlier, but does not permit additional axons to cross.

## Discussion

During the initial projection of commissural axons, Netrin/DCC signaling promotes axonal outgrowth and attraction toward the midline. Meanwhile, ROBO1/2 are inhibited by a number of factors and consequently the axons are believed to be nonresponsive to the SLIT family of midline repellents (*Figure 8*). Our previous study shows that NOVAs modulate DCC activity by regulating the production of two distinct splice variants, $DCC_{long}$ and $DCC_{short}$ (*Leggere et al., 2016*). In *Nova1/2* dKO embryos, where $DCC_{long}$ is reduced, fewer axons are able to reach the midline, similar to in *Dcc* KO embryos (*Figure 8*). In this study, we report that NOVAs also control midline repulsion by regulating the temporal production of two *Robo1/2* isoforms, e6b+ and e6b-. The ROBO1/2(e6b+) and ROBO1/2(e6b-) receptors have minute sequence variations yet display distinct activities, with e6b+ being more repulsive than e6b- (*Figure 5A,B*; *Figure 5—figure supplement 1A*). *Robo1/2(e6b+)* is initially predominantly expressed in spinal commissural neurons. Over development, as *Nova1/2* levels increase in spinal commissural neurons, e6b+ expression is reduced while e6b- expression is elevated (*Figure 7A,B*; *Figure 8*). Consequently, we found that ROBO1/2 repulsion is not uniformly inhibited precrossing. Instead, when e6b+ is predominantly expressed early on, commissural axons are sensitive to SLIT, because loss of e6b+ in *Robo1* e6bΔ/Δ mutants allows the axons to cross the midline precociously (*Figure 7C,D*). Later on, in order for axons to enter the midline, ROBO1/2 repulsion must be attenuated and sufficiently inhibited, as the persistence of e6b+ as the predominant variant in *Nova1/2* dKO embryos blocks midline crossing in some axons (*Figure 4*; *Figure 8*). Thus, besides being predominantly attracted by Netrin, precrossing axons are also somewhat repelled by SLIT to ensure proper timing in approaching and crossing the midline. A balance between Netrin and SLIT signaling systems has been implicated in other guidance events, such as in positioning spinal motor neuron cell bodies and in guiding longitudinal axons in the midbrain (*Kim et al., 2015*; *Kim et al., 2014*).

Taken together, the defects in midline crossing in *Nova1/2* dKO mutants are twofold. First, the deficiency in Netrin/DCC attraction causes fewer axons to reach the midline. Second, elevated SLIT/ROBO repulsion partially blocks axons from entering once they arrive at the midline (*Figure 8*). Although elevated ROBO repulsion may also prevent some axons from getting to the midline, the

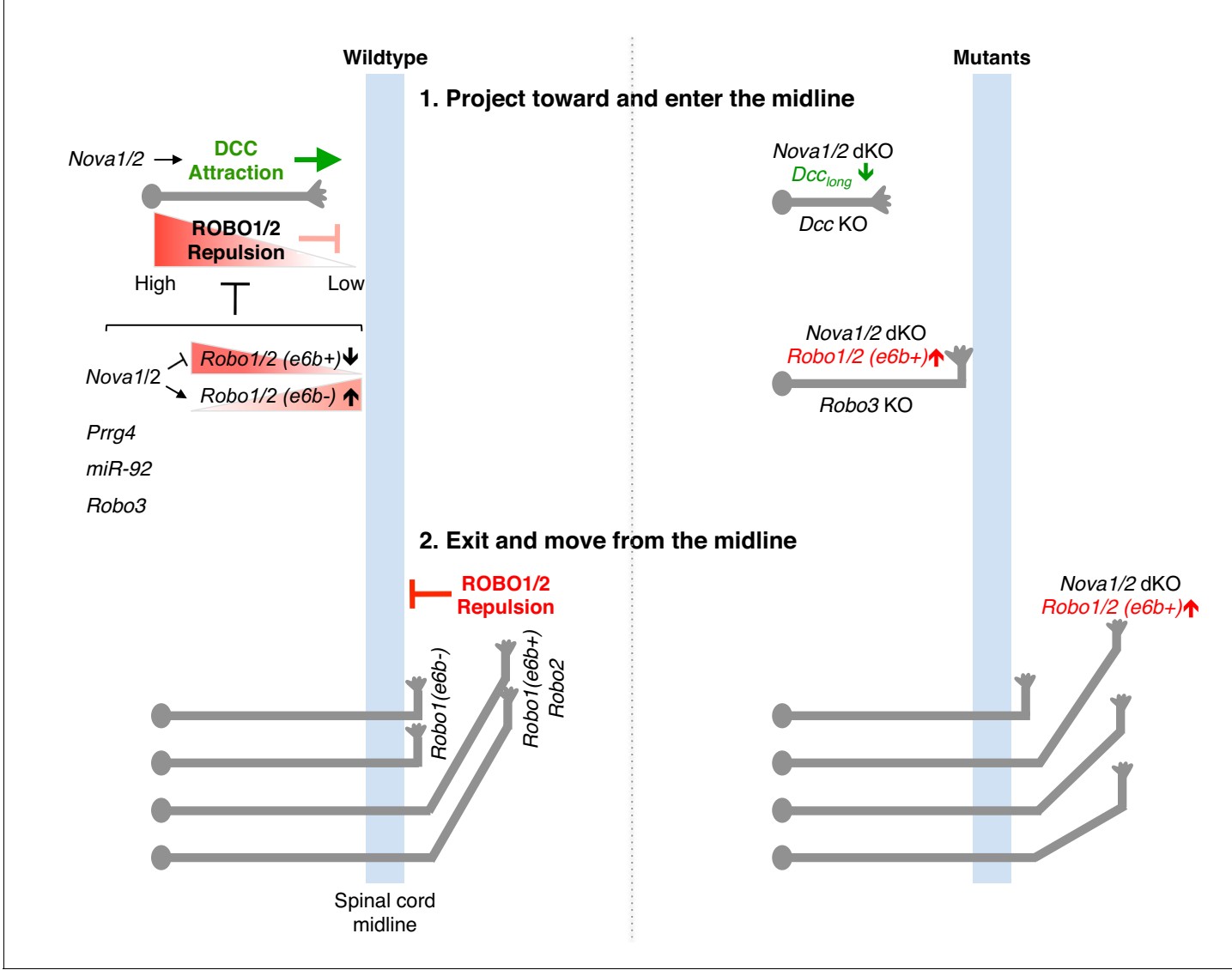

**Figure 8.** Model of NOVA1/2 regulation of commissural axon guidance. As commissural axons approach and enter the midline, Netrin/DCC signaling promotes axonal outgrowth and midline attraction. *Nova1/2* regulate *Dcc* alternative splicing by promoting the production of the full-length *Dcc$_{long}$* isoform. In *Nova1/2* dKO embryos, *Dcc$_{long}$* is reduced and fewer axons are able to reach the midline, phenocopying *Dcc* KO mutants. In precrossing axons, SLIT/ROBO repulsion is repressed by multiple genes including *Prrg4*, *miR-92*, and *Robo3* to allow midline entry. We show here that ROBO repulsion is not uniformly repressed precrossing, but is developmentally controlled such that ROBO repulsion is active early on to prevent premature crossing and is sufficiently inhibited later to allow crossing. This is achieved through the regulation of *Robo1/2* alternative splicing at microexon 6b by the NOVA splicing factors. As NOVAs increase during development, the *Robo1/2(e6b+)* isoform, which has a greater repulsive activity, is reduced, whereas the *Robo1/2(e6b-)* isoform is upregulated. In *Nova1/2* dKO embryos, e6b+ is abnormally elevated and the axons that are able to reach the midline often fail to enter. A similar and more complete defect is present in *Robo3* KO embryos and partial loss of function in *Robo3* and *Nova1/2* synergistically blocks midline entry. Upon entering the midline, SLIT/ROBO repulsion is upregulated, which facilitates axonal exit, blocks recrossing, and guides postcrossing axons to appropriate lateral positions. *Robo1(e6b-)* directs the axons medially, while *Robo1(e6b+)* and *Robo2* guide the axons dorsolaterally. Double knockout of *Nova1/2* reduces the medial tract and increases the dorsolateral tract.

DOI: https://doi.org/10.7554/eLife.46042.022

fact that restoring *Dcc$_{long}$* expression in *Nova1/2* dKO embryos almost completely reverses the defect argues that ROBO repulsion does not play a major role (***Leggere et al., 2016***). Consistent with this notion, overexpression of ROBO3.1, which represses ROBO1/2-mediated repulsion, is not sufficient to restore axonal outgrowth and projection to the midline in *Nova1/2* dKO embryos (***Leggere et al., 2016***). Similarly, dampened DCC attraction may also block some axons from

entering the midline. However, the fact that reducing *Robo1/2(e6b+)* levels largely enables crossing in *Nova1/2* dKO embryos argues against a major contribution by DCC misregulation. Thus, NOVAs regulate both midline attraction and repulsion by controlling the alternative splicing of two groups of key guidance receptors, DCC and ROBO1/2. Consistent with our findings in the spinal commissural neurons, *Dcc* and *Robo2* alternative splicing has been reported to be disrupted in the same manner in E18.5 *Nova2* KO cortex (*Saito et al., 2016*). *Nova2* KO mutants are missing the corpus callosum, a major brain commissure, and restoring *Dcc_long* expression in *Nova2* KO cortex alone is not sufficient to rescue midline crossing by callosal axons (*Saito et al., 2016*). Our results suggest that restoring the normal *Dcc_long* level and simultaneously reducing *Robo1* and/or *Robo2* e6b+ expression are likely to be required to rescue the acallosal phenotype.

Given that *Nova1/2* dKO does not significantly alter the overall levels and patterns of *Robo1/2* expression, it is likely that the ROBO1/2 isoforms are subject to similar repression in their translation and/or trafficking, such as by mir-92 and PRRG4 (*Justice et al., 2017*; *Yang et al., 2018*). As overexpressing either e6b isoform results in the presence of the receptors on precrossing axons and blocks crossing (*Figure 5*), downregulating both isoforms will be crucial to inhibiting midline repulsion. Due to a lack of specific antibodies to distinguish between the endogenous ROBO1/2 isoforms and the fact that ectopic expression alters the normal protein distributions, we could not directly test if the different ROBO1/2 isoforms are similarly inserted into the axonal surface in vivo (*Figure 5C*). Without affecting ROBO1/2 levels or localizations, both *Robo3* and *Nova1/2* inhibit ROBO1/2 receptor activities. The mechanism underlying ROBO3-mediated inhibition remains unknown, and NOVAs repress the production of the e6b+ variant with a higher repulsive activity (*Figure 3D*; *Figure 5*). Partial loss of function in *Nova1/2* and *Robo3*, which somewhat increases *Robo1/2(e6b+)* expression (*Figure 3— figure supplement 1D*) and reduces *Robo1/2* inhibition, respectively, can synergistically block midline crossing (*Figure 2C,D*). Taken together, repressing ROBO1/2 precrossing involves the collaboration of multiple mechanisms, and the microexon 6b isoforms are likely to be regulated by all the factors, although there may be some quantitative differences. The temporal regulation of exon 6b by NOVAs remains to be further examined at the protein level and at single-neuron resolution, and we cannot yet distinguish if it plays an instructive or a permissive role in midline crossing.

Given that deleting *Robo1(e6b+)*, but not *Robo2(e6b+)*, allows premature midline crossing, *Robo1*-expressing axons are likely to be pioneer axons that initiate midline crossing. *Robo2*-expressing axons, by contrast, may cross by following an existing axonal scaffold. The leader and follower commissural axons have been demonstrated to have distinct growth cone morphology and growth kinetics during midline crossing (*Bak and Fraser, 2003*). This hypothesis may help to explain why *Robo2* deletion alone cannot promote midline crossing in *Robo3* KO mutants, but can help facilitate crossing when *Robo1* is also deleted to allow some axons to cross first (*Jaworski et al., 2010*). In this study, we similarly observed that *Robo2(e6b+)* reduction alone cannot rescue the midline crossing deficit in *Nova1/2* dKO embryos, but can rescue the defect when combined with *Robo1(e6b+)* reduction (*Figure 4*).

Once axons reach the contralateral side, ROBO repulsion becomes upregulated. ROBO3.1 and mir-92 have been shown to be absent on postcrossing axons (*Chen et al., 2008*; *Yang et al., 2018*), and factors including USP33, RabGDI, and CLSTN1 stabilize or promote ROBO1 surface level postcrossing (*Alther et al., 2016*; *Philipp et al., 2012*; *Yuasa-Kawada et al., 2009*). High ROBO repulsion thus facilitates midline exit, blocks reentry, and guides postcrossing axons away from the midline (*Figure 8*). Previous studies have shown that *Robo1* directs longitudinally projecting axons close to the midline, whereas *Robo2* guides them more dorsolaterally (*Farmer et al., 2008*; *Jaworski et al., 2010*). We show here that distinct ROBO1 isoforms help position axons in both tracts, with *Robo1(e6b-)* directing axons to the medial position and *Robo1(e6b+)* guiding them more laterally (*Figure 5*). In *Nova1/2* dKO embryos, more axons join the dorsolateral tract and reducing *Robo1/2(e6b+)* expression can rescue the defect (*Figure 4*). By contrast, in the absence of *Robo1(e6b+)*, as in *Robo1* e6bΔ/Δ embryos, more axons join the medial tract (*Figure 5—figure supplement 1B*). The lateral funiculi are still present in *Robo1* e6bΔ/Δ embryos, suggesting that *Robo2* activity is sufficient to direct the axons laterally, consistent with previous findings (*Farmer et al., 2008*; *Jaworski et al., 2010*).

It is important to note that within different commissural neuron populations, DCC and ROBO1/2 can have distinct guidance activities. For example, the ventral-most spinal commissural neurons do not require Netrin to project toward the midline (*Rabe et al., 2009*). In the hindbrain, Netrin/DCC

signaling is not required to attract commissural axons toward the midline, but it instead attracts postcrossing axons close to the midline. In these hindbrain commissural axons, ROBO1/2 are important for expelling axons out of the midline, but are not involved in their postcrossing projection (*Shoja-Taheri et al., 2015*). The available tools for our analyses, including DiI and axonal markers (e.g. anti-ROBO3 and anti-L1), did not allow us to distinguish between specific subgroups of commissural axons.

As SLIT/ROBO signaling is only partially responsible for midline repulsion, additional guidance pathways, including Ephrin/Eph, Semaphorin/Neuropilin/Plexin, and Slit2(C-term)/PlexinA1 are in play (*Delloye-Bourgeois et al., 2015*; *Kullander et al., 2003*; *Zou et al., 2000*). Many of these guidance molecules have been reported to have splice variants (*Cackowski et al., 2004*; *Holmberg et al., 2000*; *Lai et al., 1999*; *Takahashi et al., 2009*; *Tamagnone et al., 1999*). Therefore, it is possible that additional alternative splicing events are involved in regulating commissural axon guidance at the midline. In fact, from our previous RNAi screen against RNA-binding proteins, we have identified additional splicing regulators with midline crossing defects (*Leggere et al., 2016*), underscoring the general importance of alternative splicing in regulating axon guidance.

## Materials and methods

**Key resources table**

| Reagent type (species) or resource | Designation | Source or reference | Identifiers | Additional information |
|---|---|---|---|---|
| Genetic reagent (*Mus musculus*) | CD-1 | Charles River Laboratories | Crl:CD-1(ICR) | |
| Genetic reagent (*Mus musculus*) | *Dcc* knockout | *Fazeli et al., 1997* | | |
| Genetic reagent (*Mus musculus*) | *Nova1* knockout | *Jensen et al., 2000* | | |
| Genetic reagent (*Mus musculus*) | *Nova2* knockout | *Saito et al., 2016* | | |
| Genetic reagent (*Mus musculus*) | *Robo1* knockout | *Long et al., 2004* | | |
| Genetic reagent (*Mus musculus*) | *Robo3* knockout | *Sabatier et al., 2004* | | |
| Genetic reagent (*Mus musculus*) | *Robo1* e6b del | This paper | | *Figure 4—figure supplement 1* |
| Genetic reagent (*Mus musculus*) | *Robo2* e6b del | This paper | | *Figure 4—figure supplement 1* |
| Genetic reagent (*Mus musculus*) | *Robo1; Robo2* e6b del | This paper | | *Figure 4—figure supplement 1* |
| Genetic reagent (*Gallus gallus*) | Chicken embryos | Charles River Laboratories | 10100326 | Specific pathogen -free fertile eggs |
| Genetic reagent (*Rattus norvegicus*) | Sprague Dawley rats | Charles River Laboratories | Crl:SD | |
| Cell line (*Cercopithecus aethiops*) | COS-1 | American Type Culture Collection | CRL-1650 | Free of mycoplasma contamination |
| Recombinant DNA reagent | *Robo1 (e6b-)* | This paper | | HA tag at C-term; pCAGGS vector |
| Recombinant DNA reagent | *Robo1 (e6b+)* | This paper | | HA tag at C-term; pCAGGS vector |
| Recombinant DNA reagent | *Robo2 (e6b-; e21-)* | This paper | | HA tag at C-term; pCAGGS vector |
| Recombinant DNA reagent | *Robo2 (e6b-; e21+)* | This paper | | HA tag at C-term; pCAGGS vector |
| Recombinant DNA reagent | *Robo2 (e6b+; e21-)* | This paper | | HA tag at C-term; pCAGGS vector |

*Continued on next page*

*Continued*

| Reagent type (species) or resource | Designation | Source or reference | Identifiers | Additional information |
|---|---|---|---|---|
| Recombinant DNA reagent | *Robo2 (e6b+; e21+)* | This paper | | HA tag at C-term; pCAGGS vector |
| Recombinant DNA reagent | *Robo1* exon 6b splicing reporter | This paper | | *Figure 3*; *Figure 3—figure supplement 1E* |
| Recombinant DNA reagent | *Robo2* exon 6b splicing reporter | This paper | | *Figure 3*; *Figure 3—figure supplement 1E* |
| Recombinant DNA reagent | *Nova1-V5* | *Leggere et al., 2016* | | V5 tag at C-term; pCAGGS vector |
| Recombinant DNA reagent | *Nova2-V5* | *Leggere et al., 2016* | | V5 tag at C-term; pCAGGS vector |
| Antibody | ROBO3 rabbit polyclonal | *Sabatier et al., 2004* | | IHC (1:1000) |
| Antibody | L1 rat monoclonal | MilliporeSigma | MAB5272 | IHC (1:1000) |
| Antibody | ROBO1 rabbit polyclonal | Novus Biologicals | NBP2-20195 | IHC (1:1000); Western blot (1:500) |
| Antibody | ROBO2 rabbit polyclonal | Novus Biologicals | NBP1-81399 | IHC (1:1000) |
| Antibody | ROBO2 mouse monoclonal | R and D Systems | MAB3147 | Western blot (1:500) |
| Antibody | TAG1 mouse monoclonal IgM | Developmental Studies Hybridoma Bank | 4D7 | IHC (1:1000) |
| Antibody | HA rat monoclonal | Roche | 3F10 | IHC (1:1000); Western blotting (1:1000, HRP conjugated) |

## Mice

*Dcc, Nova1, Nova2, Robo1*, and *Robo3* KO mutants were generated and characterized previously (*Fazeli et al., 1997*; *Jensen et al., 2000*; *Long et al., 2004*; *Sabatier et al., 2004*; *Saito et al., 2016*). All mouse strains were outcrossed to the CD-1 mouse strain. Heterozygous animals were interbred to generate WT and KO animals. Timed pregnant Sprague Dawley rats were purchased from Charles River to obtain embryos of different developmental stages.

## CRISPR/Cas9 knockout mice

*Robo1* e6bΔ, *Robo2* e6bΔ, and *Robo1* e6bΔ; *Robo2* e6bΔ mutants were generated using the CRISPR/Cas9 technology (*Hsu et al., 2014*). Single guide RNAs were selected using tools developed by the Zhang laboratory at MIT (website crispr.mit.edu; algorithm described in *Hsu et al., 2013*). *Cas9* mRNA and single guide RNA were in vitro transcribed using mMESSAGE mMACHINE T7 kit and MEGAshortscript T7 kit (Thermo Fisher Scientific), respectively. RNA products were purified using MEGAclear kit (Thermo Fisher Scientific) following the manufacturer's protocol. RNAs were injected into pronuclei at 100 ng/µl for *Cas9* mRNA and at 25 ng/µl for each single guide RNA. Two guide RNAs that flank exon 6b were injected to introduce genomic DNA deletions in *Robo1* or *Robo2* alone. Simultaneous injection of four guide RNAs was used to delete exons 6b from both *Robo1* and *Robo2*. Animals harboring desired deletions were selected by genomic DNA PCR reactions and confirmed by Sanger sequencing. Mutant mice were outcrossed to CD-1 mice and were

used to breed with *Nova1/2* dKO mice to generate compound mutants. Homozygous *Robo1/2* exon 6b deletion mutants are viable and fertile.

**Single guide RNAs used to generate exon 6b deletions in *Robo1/2* (also see *Figure 4—figure supplement 1A*).**

| Allele | Robo1 | Robo2 |
|---|---|---|
| *Robo1* e6bΔ | Intron6: GAGTCTTGAAATCGATACTAtgg (PAM sequence in lower case) Intron6b: ACTGCACAGAATAAATCTGCagg | - |
| *Robo2* e6bΔ | - | Intron6: TCATAATTCAGTTATGAATAagg Intron6b: ACTAAAGCGACCGAAAAGCCagg |
| *Robo1* e6bΔ; *Robo2* e6bΔ | Intron6: GAGTCTTGAAATCGATACTAtgg Intron6b: ACTGCACAGAATAAATCTGCagg | Intron6: CTGTAGACATTACAATGGTGtgg Intron6b: ACTAAAGCGACCGAAAAGCCagg |

## cDNAs

Total RNA was isolated from E11.5 CD-1 mouse spinal cord, and *Robo1/2* cDNAs were cloned by RT-PCR and confirmed by Sanger sequencing. For alternatively spliced sequences in *Robo1/2* that were not affected by *Nova1/2* dKO, the most abundant variant was used for the cDNAs. An HA tag was added to the C-terminus to help detect protein expression. *Nova* cDNAs used in the splicing assays were previously described (*Leggere et al., 2016*).

## Alternative exon numbering

Exon numbers were designated using NCBI reference sequences NM_019413.2 for *Robo1* and NM_175549.4 for *Robo2*. Exon 6b from either gene is not included in the reference sequence and is located between exons 6 and 7 of the reference sequence.

## DiI tracing

DiI labeling of spinal commissural axons was carried out as previously described (*Chen et al., 2008*). E12.5 spinal cords were dissected and fixed with 4% paraformaldehyde (PFA). Vybrant DiI cell-labeling solution (Thermo Fisher Scientific) was microinjected into one side of the spinal cord at mediolateral positions. Spinal cords were incubated overnight at 37°C in 1xPBS, prepared in an openbook configuration, and imaged with fluorescence microscopy.

For confocal microscopy, images of openbook spinal cord preparations were acquired on a Zeiss 510 LSM confocal microscope using ZEN imaging software. For each image, 20–24 3.0 µm thick optical sections were acquired. Maximum intensity Z-projections were then generated from 20 consecutive optical sections. All images were processed using Fiji software.

## Immunohistochemistry (IHC)

IHC was carried out as previously described (*Xu et al., 2014*) using the following antibodies: anti-L1 (MAB5272, MilliporeSigma), anti-ROBO1 (NBP2-20195, Novus Biologicals), anti-ROBO2 (NBP1-81399, Novus Biologicals), anti-ROBO3 (*Sabatier et al., 2004*), anti-TAG1 (4D7, Developmental Studies Hybridoma Bank), and Alexa Fluor 594-conjugated secondary antibodies (Jackson ImmunoResearch). The antibodies were used at a final concentration of 0.5–1 µg/ml.

## In ovo electroporation

Culturing of chicken embryos and in ovo electroporation were performed as described previously (*Chen et al., 2008*). *Actb-gfp* (*βactin-gfp*) and *Robo1/2* expression constructs (in the pCAGGS vector) were microinjected into the neural tube of E3 chicken embryos (stage 18) and electroporated into one half of the spinal cord (ECM830 electroporator, 30 volts, 50 ms/pulse, five pulses). The concentration of the plasmids was 150 ng/µl. After culturing for an additional 48 hr at 37°C (until stage 25), spinal cords were microdissected, fixed with 4% PFA, and imaged in an openbook configuration with fluorescence microscopy. Commissural axons were distinguished from ipsilateral axons as the former turned longitudinally next to the midline while the latter turned at a more dorsolateral

position. To assess exogenously expressed mouse ROBO proteins, anti-HA (3F10, Roche) was used to stain transverse sections of chicken spinal cord. In addition, we lysed chicken spinal cords with lysis buffer (10 mM Tris, 150 mM NaCl, 1% Triton X-100, pH 8, with protease inhibitors). After centrifugation to remove undissolved materials, SDS loading buffer (4x with 200 mM Tris-HCl pH6.8, 8% SDS, 0.4% bromophenol blue, 400 mM DTT) was added to the supernatant and the lysate was analyzed by SDS-PAGE using anti-HA conjugated with HRP (Roche).

## Whole embryo culture

Whole embryo culture was carried out as previously described (*Chen et al., 2008*). Embryos were electroporated at E9.75 with *Actb-gfp* (*βactin-gfp*) and cDNAs into one side of the spinal cord and were cultured for 40 hr at 37°C. Spinal cords were microdissected, fixed with 4% PFA, and imaged in an openbook configuration with fluorescence microscopy. We have previously shown that *Actb-gfp* (*βactin-gfp*)-expressing axons are almost exclusively ROBO3-positive commissural axons due to the developmental stage at the time of injection (*Leggere et al., 2016*).

## Quantification of phenotypes

For quantifying midline crossing from DiI labeling, fluorescent axonal signals from the contralateral side were compared to those from the ipsilateral axons, as previously described (*Chen et al., 2008*). 3–5 DiI injection points were analyzed for each animal. The signals were measured using ImageJ (NIH). For quantifying *gfp*-labeled axon midline crossing, the same measurements were taken and compared, except that areas with cell bodies were avoided to reduce background.

For quantification of axon lateral positioning in chicken embryos, the distance between the dorso-lateral-most axons and the midline was compared to the total dorsal-ventral height of the spinal cord (distances 1 and 2 in *Figure 5A*, respectively). For quantifying the lateral positioning of anti-L1 or anti-ROBO1 labeled axons, the thickness of the lateral and ventral funiculi were measured using ImageJ, as previously described (*Jaworski et al., 2010*). As the number of postcrossing axons was reduced in *Nova1/2* dKO embryos due to fewer axons reaching the midline, we compared the thickness of the lateral and ventral funiculi from the same section (see *Figure 4C*). 5–10 sections from each animal were analyzed. As it is difficult to obtain a large number of compound mutants, we used the rostral half of the spinal cord for DiI labeling and the lumbar level for anti-L1 and anti-ROBO1 staining. To reduce variation in developmental stages, littermates of comparable sizes and from more than three litters were examined.

For quantifying the ventral commissure size in *Figures 2A* and *7C*, the thickness of the commissure was compared to that of the floor plate. Measurements were taken using ImageJ. To minimize developmental variations, we examined animals from at least three different litters, selected embryos of the same size, and used anti-TAG1 patterns to further gauge the developmental stages (see *Figure 7—figure supplement 1*). 5–10 spinal cord sections from each animal were quantified.

## Quantitative and semi-quantitative RT-PCR

Spinal cord tissues were microdissected, and the dorsal and ventral halves were separated to distinguish between the commissural and motor neuron populations. Rat tissues were used to examine temporal gene/exon expression due to the ease of microdissection at earlier stages. Total RNA was extracted using Trizol (Thermo Fisher Scientific), and reverse transcription was carried out using Maxima RT (Thermo Fisher Scientific). Quantitative PCR was carried out using a Realplex$^2$ thermocycler (Eppendorf). Semi-quantitative PCR was performed to generate multiple isoforms in a single reaction and to compare the relative expression by electrophoresis. The cycle number used in semi-quantitative PCR was determined by quantitative PCR to obtain products during the exponential amplification phase.

**Primers used for quantitative PCR**

| Gene | Amplicon | Forward primer | Reverse primer |
|------|----------|----------------|----------------|
| *Robo1* | E6b- | gacagttcaagagccgccacattt | gatttccagttgcttcgcactg |
| | E6b+ | cgctactttgacagttcaagttggg | gatttccagttgcttcgcactg |

*Continued on next page*

*Continued*

**Primers used for quantitative PCR**

| Gene | Amplicon | Forward primer | Reverse primer |
|---|---|---|---|
|  | Common exons | gggatcatacacttgtgtggcagaa | gatttccagttgcttcgcactg |
| Robo2 | E6b- | ccctcactgtccgagctcctcc | ttgagcaacgatctgatctcttgg |
|  | E6b+ | ccgagttcgccctgttgctcc | ttgagcaacgatctgatctcttgg |
|  | E21- | ggaacaacggtgggaaaggtgg | ggaggaggaggtaga |
|  | E21+ | gcaccaccagctctcacaacagc | ggaggaggaggtaga |
|  | Common exons | agtggaagcctctgctaccctc | ttgagcaacgatctgatctcttgg |
| Nova1 | Common exons | ctcgcggaaaaggccgcttg | gtactggccgtcttcgcccgt |
| Nova2 | Common exons | cgacagagccaagcaggcca | acggtcaccacgcgctcttg |
| Hprt | Common exons | tgacactggtaaaacaatgca | tcaaatccaacaaagtctg |

**Primers used for semi-quantitative PCR (amplifying multiple isoforms)**

| Gene | Amplicon | Forward primer | Reverse primer |
|---|---|---|---|
| Robo1 | Exon 6b | gggatcatacacttgtgtggcagaa | ctggtcccgaggttttacaacg |
|  | Exon 18 | gcaagaagagaaacggactcacca | cggcctccctccactgctg |
|  | Exon 21 | ccatggctggcagacacg | ctggatgagttgagtggtggc |
| Robo2 | Exon 6b | agtggaagcctctgctaccctc | ttgagcaacgatctgatctcttgg |
|  | Exon 21 | ccacagtggaaaagctcagttca | ggaggaggaggtaga |
|  | Exon 24b | cccaggcccctcagagcacta | gtgggccgctgcctttgaga |
|  | Exon 26 | acagccagtgttacctcatcgg | ctgatgagctgtgcccgcca |
| Foxp1 | Exons 15 to 17 of *Foxp1* with exons 16 and 16b sequences replaced by *Robo1/2* exon 6b (also see below and *Figure 3—figure supplement 1E*). | gaatgtttgcttacttccgacgc | agtaggcgtggctgctctgc |

## Splicing assay

The total genomic sequence between exons 6 and 7 is 25 kb in *Robo1* and *Robo2*, with candidate NOVA-binding sites located in flanking introns adjacent to exon 6b. We cloned an approximately 500 bp genomic fragment containing exon 6b and the NOVA-binding sites, and inserted it in between *Foxp1* exons 15 and 17 to replace exons 16 and 16b. *Foxp1* exons 15 and 17 are constitutive, whereas exons 16 and 16b are alternatively spliced under the control of *Nova* (*Zhang et al., 2010*). In the exon 6b splicing reporter, *Foxp1* exons 16/16b and the surrounding NOVA-binding sites were deleted and replaced by *Robo1/2* exon 6b and its flanking intron sequences (see *Figure 3—figure supplement 1E*). The *Foxp1* reporter construct is in the pcDNA3 backbone and was provided by the Darnell group at Rockefeller University. We transfected COS-1 cells with the splicing reporter with TransIT-LT1 (Mirus Bio), together with *Nova1*, *Nova2*, or an empty expression vector. The cells were cultured for 48 hr and the total RNA was collected using Trizol (Thermo Fisher Scientific). Reverse transcription was performed from a BGH reverse primer (5'aaacaacagatggctggcaact3') using SMARTScribe reverse transcriptase (Clontech), and semi-quantitative PCR was performed to amplify splice isoforms. A V5 tag at the C-terminus of NOVA1 and NOVA2 was used to confirm protein expression by western blotting. Point mutations were introduced by PCR reactions using Pfu polymerase (Agilent), and were confirmed by Sanger sequencing.

To generate the *Robo2* exon 21 splicing reporter, we cloned the full genomic DNA between exons 20 and 22 (5.8 kb total) into the pcDNA3.1 vector containing a CMV promoter. Splicing assays with WT and mutant binding sites were performed in the same way as described above.

## Small GTPase activation assay

COS-1 cells were grown in 24-well plates and transfected in reduced serum medium Opti-MEM (Thermo Fisher Scientific) for 24 hr. Plasmids were used at 100 ng/well for *Robo1/2* and 50 ng/well for *Cdc42/Rac1*. We found that higher amounts of *Robo* cDNA or longer periods of expression can affect CDC42/RAC1 activation in the absence of SLIT2 stimulation. Cells were stimulated with 500 ng/ml mouse SLIT2 N-term (R and D 5444-SL) for 10 min at 37°C, and were lysed on ice for 5 min in 50 mM Tris, pH7.5, 10 mM $MgCl_2$, 150 mM NaCl, 2% IGEPAL, and protease inhibitors (Roche). After centrifugation at 14,000 rpm for 5 min at 4°C, 1/10 of the supernatant was kept to detect ROBO1/2 and total CDC42/RAC1. To the rest of the supernatant, EDTA was added at 15 mM and GTPγS at 0.2 mM. After incubation at room temperature for 15 min, $MgCl_2$ was added at 60 mM to stop GTPγS loading. PAK-GST beads (Cytoskeleton), which specifically bind GTP-bound CDC42/RAC1, were added and the samples were incubated with rotation at 4°C for 1 hr. After three washes with 50 mM Tris, pH7.5, 30 mM $MgCl_2$, and 150 mM NaCl at 4°C, bead-bound proteins were analyzed by western blotting. ROBO1/2 proteins were detected by a C-terminal HA peptide tag (HRP-conjugated anti-HA, 3F10, Roche) and CDC42/RAC1 were detected by a C-terminal FLAG tag (HRP-conjugated anti-FLAG, M2, Sigma Aldrich). To quantify the fold change in CDC42/RAC1 activation, the signal intensity of protein bands was measured using ImageJ. GTP-bound protein was first normalized to the total protein for each condition. The ratio between the normalized signals from before and after SLIT2 stimulation was then calculated as the fold change for each ROBO1/2 isoform. Three independent assays were performed and are illustrated in the graphs in *Figure 6B*.

## AP-SLIT2/ROBO-binding assay

AP-SLIT2 was produced by fusing alkaline phosphatase to the N-terminus of human SLIT2. This fusion protein has been previously shown to bind the ROBO receptors (*Brose et al., 1999*). The fusion protein was produced in HEK293T cells and the protein concentration was determined by measuring the AP activity against an AP protein standard (Colorimetric AP assay kit, Abcam). COS-1 cells were transfected with *Robo1/2* cDNAs and cultured for 24 hr. Binding of AP-SLIT2 was performed using a previously described protocol (*Cheng and Flanagan, 2001*). Briefly, AP-SLIT2 proteins were added to the culture medium at a series of concentrations and binding was performed at room temperature for 90 min in the presence of 1 µg/ml heparin. After 3–6 washes in Hank's balance buffer supplemented with 0.5 mg/ml BSA, 20 mM HEPES, and 1 µg/ml heparin, pH 7.4, the cells were lysed (10 mM Tris, 150 mM NaCl, 1% Triton X-100, pH 8, with protease inhibitors). After clearing the lysate by centrifugation, AP substrate, para-nitrophenyl phosphate, was added to the lysate and optical density was measured at 405 nm. Data were analyzed using Prism7 and fitted with one site specific binding with Hill slope method. Three independent binding assays were performed and the level of binding was normalized to the saturating level of signaling.

## Cell surface biotinylation assay

Biotinylation was performed as described previously (*Lai et al., 2017*). Briefly, *Robo1/2* cDNAs were transfected into COS-1 cells. At 24 hr post-transfection, the cells were washed with cold 1x PBS (pH 8.0), and biotinylated at 4°C with EZ-Link-Sulfo-NHS-SS-Biotin (Thermo Fisher Scientific) following the manufacturer's protocol. After removing excess reagent and washing the cells with 1xPBS, the cells were lysed (10 mM Tris, 150 mM NaCl, 1% Triton X-100, pH 8, with protease inhibitors). After clearing the lysate by centrifugation, biotinylated cell-surface proteins were pulled down using NeutrAvidin agarose (Thermo Fisher Scientific). Proteins in the total lysate and in the pull-down fraction were analyzed by SDS-PAGE and western blotting.

## Acknowledgements

This work was supported by grants: Boettcher Foundation (ZC); Linda Crnic Institute (ZC); National Institutes of Health (NIH) R01EY024261 (HJJ). We thank Yuhki Saito and Robert B Darnell for reagents, Yudong Teng at the MCDB transgenic facility for generating knockout mice, Caleb Anderson, Maxwell L Saal, and Kelsey R Arbogast for technical assistance, Lee Niswander for critical comments, and Aileen Sewell and Heidi Chial for editing.

# Additional information

## Funding

| Funder | Grant reference number | Author |
| --- | --- | --- |
| Boettcher Foundation | | Zhe Chen |
| National Institutes of Health | R01EY024261 | Harald J Junge |
| Linda Crnic Institute for Down Syndrome | | Zhe Chen |

The funders had no role in study design, data collection and interpretation, or the decision to submit the work for publication.

## Author contributions

Verity Johnson, Data curation, Methodology, Writing—review and editing; Harald J Junge, Resources, Funding acquisition, Writing—review and editing; Zhe Chen, Conceptualization, Data curation, Formal analysis, Supervision, Funding acquisition, Validation, Methodology, Writing—original draft, Project administration, Writing—review and editing

## Author ORCIDs

Harald J Junge https://orcid.org/0000-0003-2458-6010
Zhe Chen https://orcid.org/0000-0003-0683-9491

## Ethics

Animal experimentation: All experimental manipulations and care of animals have been approved by the University of Colorado Boulder Institutional Animal Care and Use Committee (protocol number 2497).

## Decision letter and Author response

Decision letter https://doi.org/10.7554/eLife.46042.025
Author response https://doi.org/10.7554/eLife.46042.026

# Additional files

## Supplementary files

• Transparent reporting form
DOI: https://doi.org/10.7554/eLife.46042.023

## Data availability

All data generated or analysed during this study are included in the manuscript and supporting files.

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
