## [Decision Letter]

Thank you for submitting your article "Alternative splicing of mammalian ROBO1/2 microexons provides temporal regulation of axonal repulsion" for consideration by *eLife*. Your article has been reviewed by Marianne Bronner as the Senior Editor, a Reviewing Editor, and three reviewers. The following individuals involved in review of your submission have agreed to reveal their identity: Grant Mastick (Reviewer #2).

The reviewers have discussed the reviews with one another and the Reviewing Editor has drafted this decision to help you prepare a revised submission.

Summary:

The three reviewers generally had positive assessments of your study on a new role for the Nova RNA binding proteins in the regulation of Robo receptor alternative splicing during spinal commissural axon guidance. They applauded your efforts to test the function of the poorly understood Robo 1 and 2 isoforms, with strong genetic experiments, expression of altered Robo constructs, and the mechanisms of splicing by the Nova1/2 splicing factors, including the timing of splicing. You show that splicing isoforms have distinct guidance and signaling activities, revealing physiological function of alternatively spliced microexons. Especially interesting is your Crispr-based manipulation of the Robo locus to restore the balance of alternative splicing in the Nova1/2 mutant background results in a significant rescue of the Nova1/2 crossing defects.

Nonetheless, the reviewers call for revision, both additional experimental analysis and textual amendments in the Discussion section:

Essential revisions:

I) Further analysisa) There was concern that the most significant function of the ex6b is shown in the context of Nova1/2 double knock-outs in which hundreds of other mRNA transcripts are also potentially changed), and that there is no definitive insight into mechanism of how the microexon impacts Robo1/2 function. To confirm that the microexon mutant mice indeed differ from wild-type only in the single microexon, the reviewers encourage you to survey additional splicing events surrounding the deleted exon. You report no changes in exon 21 and 26 – but generally the concern in such mutants is the unintentional generation of cryptic splice acceptor or donor sites in the mutant transcript in proximity to the mutated region. One question is whether splice isoforms are expressed at comparable levels in your gain of function experiments. Given the subtle nature of the protein coding differences in these Robo isoforms, it is important to demonstrate that they are comparably expressed in the context of the guidance events (not just in heterologous cells).

b) The reviewers ask you to probe with quantitative methods whether there are any changes in total Robo1/2 mRNAs or proteins in the mutants, because the staining in Figure 4—figure supplement 2 are not quantitative and the other functional experiments involve overexpression, which are difficult to interpret.

Thus, we ask that you to assess RNA and protein levels, as well as splice junctions, in NOVA1/2 DKO mice, to confirm whether the manipulations caused an indirect effect on protein or RNA stability. The outcome could change the interpretation of your results but may also strengthen your conclusions on splice isoforms.

II) Textual additionsa) Figure 1 shows that Nova1/2 mutants have a clear change in post-crossing trajectories, causing axons to angle away from the midline. While this may reflect changes in Robo-mediated repulsion, the text should consider the possibility that the post-crossing errors caused by Nova1/2 mutations also alter Netrin guidance. A study by Shoja-Taheri, et al., 2015 showed similar post-crossing errors in Netrin1 mutant hindbrain commissural axons, and explant experiments provided support for the retention of post-crossing commissural axon responsiveness to Netrin midline attraction. Given the similarity in axon responses, and your previous evidence of Nova1/2 regulation of Netrin/DCC signaling, the Shoja-Taheri paper should be cited and considered in interpreting the post-crossing trajectories.

b) Another reviewer request is that the Discussion section would be improved by integrating the Robo1/2 splicing evidence with the other previously identified mechanisms for Robo1/2 regulation. How do you weigh the relative importance of the other levels of regulation, such as receptor localization, interaction with other receptors and other signaling systems, etc. An obvious question to raise is to what extent that the regulatory steps may interact or be dependent on each other: are they hierarchical, or additive? Could the splice isoforms be differentially regulated by receptor trafficking mechanisms, for example?

c). A major issue is the conflict between the present results and those of your previous study -i.e., that both DCC and Robo splicing can rescue the Nova1/2 DKO phenotype. Previous work from your lab has implicated Nova1 and 2 in the control of alternative splicing of the Netrin receptor DCC (2016).

You show that DCC and Nova mutants result in strikingly similar phenotypes with similar severity in terms of midline crossing defects. Rescue experiments indicating that expression of specific Nova-dependent splice variants of DCC can restore midline crossing in Nova mutants, together with a number of in vitro assays, argue that Nova and DCC function together to regulate axon growth across the midline. In the present paper, you begin by showing that Nova1/2, DCC triple mutants have even stronger defects than DCC or Nova alone, suggesting that Nova and DCC act in parallel (Figure 1).

Moreover, why should both over-expressing DCC, and reducing Robo lead to a suppression of the Nova phenotypes? This is particularly puzzling given that in your prior work, you found that in contrast to expression of DCC (which rescues Nova mutants) that expression of Robo3 (which should attenuate Robo repulsion to some degree) has no effect (2016, Figure 8). In contrast, in the current work, you detect trans-heterozygous interactions between Robo3 and the Nova genes that reveal striking LOF phenotypes and suggest that Nova functions with Robo3 (Figure 2).

The reviewers thus urge you to reconcile this apparent disconnect in the Discussion section, especially the new genetic interactions with Robo3, since your previous study argued for the specificity of Nova function with DCC and showed that Robo3 was unable to rescue Nova phenotypes. You should also consider including DCC in your current model in Figure 8.

[Editors' note: further revisions were requested prior to acceptance, as described below.]

The manuscript has been improved and the reviewers thought your amendments were appropriate and sufficient to move forward. They continue to be concerned that the read-outs for phenotypes from dye tracing or antibody labeling to indicate bundle thickness, etc. are semi-quantitative, and that measurements of Robo1/2 mRNA and protein levels are limited by analysis of whole spinal cord rather than relevant cell populations within the spinal cord. In addition, questions remain about the differential action of the 6b+ vs 6b- isoforms. Nonetheless, they feel that the study does highlight a critical function of a micro exon in vivo.

However, there are some remaining issues that need to be addressed before acceptance, as outlined below. As Reviewing Editor, I found that the Abstract, Discussion section and model in Figure 8 need further revision; the revisions listed would make the results more accessible and exhibit your story in the best light.

1) Your summary of results in the Introduction is very clear: "Here we report that Robo1/2 alternative splicing at microexon 6b is crucial for axon guidance and is controlled by the NOVA (Neuro-oncological ventral antigen) family of splicing factors…. We show that loss of Nova1/2 alters the expression of exon 6b and leads to severe midline crossing and postcrossing guidance defects. Genetically restoring the expression profile of Robo1/2 exon 6b is able to reverse these defects in Nova mutants. Interestingly, exon 6b alternative isoforms display distinct guidance activities and their production is developmentally regulated. Consequently, ROBO-mediated repulsion is not uniformly repressed precrossing as previously believed, but is instead activated initially to block premature crossing and is sufficiently blocked during crossing."

The Abstract, however, is very vague and does not inform the reader on the results. You do not indicate that microexon 6b splice variants of Robo1/2 are controlled by Nova1/2. The abstract should be rewritten to be more precise and reflective of the results.

2) Likewise, the beginning of the Discussion section should summarize the results more concisely, as in the paragraph at the end of the Introduction. The remainder of the Discussion section wanders and there is a degree of hand-waving. Your findings and the advances they make in the control of DCC-Netrin and Robo-Slit signaling should be stated more clearly.

3) The model. In Figure 8, though very welcome, is not graphically clear; the reader should be able to glean the results without looking at the legend.

a) "Mutant" should be "Mutants", as more than one mutant is included.

b) 1. and 2. titles should either be put on the midline (so as to refer to both Wildtype and Mutants) or the font should be bigger

c) Robo1/2 inhibition in red, and "low repulsion" are confusing. Put a space between axon under DCC and Robo1/2.

d) The graphic for Microexon 6b alternative splicing being developmentally controlled is not obvious.

4) Figure 8 legend is not at all clear; it attempts to compare wildtype and mutant but does not indicate wildtype vs mutant.

a) For example, Figure 8 legend: "Nova1/2 regulate (should be "regulates") Dcc alternative splicing (of what?), and (should be: "but in the Nova1/2 dKO", Nova deficiency reduces Dcc_long_ and causes fewer axons to reach the midline".

b) On the right of the diagram under Mutant, as drawn, it is not clear whether premature turning occurs with Robo3 KO or Nova1,2 dKO (when Robo1/2 (e6b+ increases), or whether there might be an interaction between robo3 and Robo1/2 and e6b+.

5) Text in English. We appreciate that you have utilized an English-speaking colleague to help edit the manuscript. However,

a) There are still many instances in this revision of inappropriate English usage, as in subsection “Microexon 6b splice variants of ROBO1/2 have distinct guidance activities”, Discussion section – "Consistently", should be "Consistent with these findings…" or similar.

Figure 5 legend – "orientated" to the left should be "oriented".

b) Throughout, there is incorrect use of singular vs plural.

---

## [Author Response]

Summary:The three reviewers generally had positive assessments of your study on a new role for the Nova RNA binding proteins in the regulation of Robo receptor alternative splicing during spinal commissural axon guidance. […] Nonetheless, the reviewers call for revision, both additional experimental analysis and textual amendments in the Discussion section:Essential revisions:I) Further analysisa) There was concern that the most significant function of the ex6b is shown in the context of Nova1/2 double knock-outs in which hundreds of other mRNA transcripts are also potentially changed), and that there is no definitive insight into mechanism of how the microexon impacts Robo1/2 function. To confirm that the microexon mutant mice indeed differ from wild-type only in the single microexon, the reviewers encourage you to survey additional splicing events surrounding the deleted exon. You report no changes in exon 21 and 26 – but generally the concern in such mutants is the unintentional generation of cryptic splice acceptor or donor sites in the mutant transcript in proximity to the mutated region. One question is whether splice isoforms are expressed at comparable levels in your gain of function experiments. Given the subtle nature of the protein coding differences in these Robo isoforms, it is important to demonstrate that they are comparably expressed in the context of the guidance events (not just in heterologous cells).

To survey alternative splicing events surrounding the deleted exon 6b, we performed RT-PCR to amplify the coding sequences between exons 5 and 8, which encode the Ig3 to Ig4 domains. To detect potential sequence changes, we cloned the RT-PCR products and Sanger sequenced 15 individual clones from the mutants. Only one product was detected, which was the e6b- isoform. The results are now included in the text and in Figure 4—figure supplement 1D.

To confirm that *Robo1/2* isoforms are comparably expressed within axons in the gain of function assays, we performed immunohistochemistry on spinal cord sections and found that different isoforms were overexpressed to similar levels and the overexpressed proteins could be detected on both pre- and post-crossing axons. The results are now included in Figure 5C.

b) The reviewers ask you to probe with quantitative methods whether there are any changes in total Robo1/2 mRNAs or proteins in the mutants, because the staining in Figure 4—figure supplement 2 are not quantitative and the other functional experiments involve overexpression, which are difficult to interpret.Thus, we ask that you to assess RNA and protein levels, as well as splice junctions, in NOVA1/2 DKO mice, to confirm whether the manipulations caused an indirect effect on protein or RNA stability. The outcome could change the interpretation of your results but may also strengthen your conclusions on splice isoforms.

Using quantitative RT-PCR and western blotting, we assessed the total mRNA and proteins levels of Robo1/2, respectively, in Nova1/2 dKO. We found that the levels were comparable to those in WT. We also cloned and sequenced the RT-PCR products from exons 5-8 in Nova dKO and confirmed that there was no cryptic splice site being used in Nova dKO. Using quantitative RT-PCR and western blotting, we also examined the total mRNA and protein levels in exon 6b deletion mutants and did not detect any significant changes from WT levels. The results are included in Figure 3—figure supplement 1A,B and in Figure 4—figure supplement 1B,C.

II) Textual additionsa) Figure 1 shows that Nova1/2 mutants have a clear change in post-crossing trajectories, causing axons to angle away from the midline. While this may reflect changes in Robo-mediated repulsion, the text should consider the possibility that the post-crossing errors caused by Nova1/2 mutations also alter Netrin guidance. A study by Shoja-Taheri, et al., 2015 showed similar post-crossing errors in Netrin1 mutant hindbrain commissural axons, and explant experiments provided support for the retention of post-crossing commissural axon responsiveness to Netrin midline attraction. Given the similarity in axon responses, and your previous evidence of Nova1/2 regulation of Netrin/DCC signaling, the Shoja-Taheri paper should be cited and considered in interpreting the post-crossing trajectories.

The previous finding by Shoja-Taheri et al., is indeed crucial for our interpretation. The Shoja-Taheri paper shows that Netrin signaling regulates post-crossing projection of commissural axons in the hindbrain, but not in the spinal cord. Our observation in the spinal cord is consistent with these findings. We have modified the text to clarify Netrin’s role in the postcrossing errors in Nova1/2 mutants.

b) Another reviewer request is that the Discussion section would be improved by integrating the Robo1/2 splicing evidence with the other previously identified mechanisms for Robo1/2 regulation. How do you weigh the relative importance of the other levels of regulation, such as receptor localization, interaction with other receptors and other signaling systems, etc. An obvious question to raise is to what extent that the regulatory steps may interact or be dependent on each other: are they hierarchical, or additive? Could the splice isoforms be differentially regulated by receptor trafficking mechanisms, for example?

We have significantly revised the text. We have discussed how ROBO splice isoforms may be regulated at different levels, such as in trafficking. We have discussed how ROBO isoforms may interact with other signaling pathways. We have also included discussions on the relationship between midline attraction and repulsion and the role of NOVA in both events.

c) A major issue is the conflict between the present results and those of your previous study -i.e., that both DCC and Robo splicing can rescue the Nova1/2 DKO phenotype. Previous work from your lab has implicated Nova1 and 2 in the control of alternative splicing of the Netrin receptor DCC (2016).
*You show that DCC and Nova mutants result in strikingly similar phenotypes with similar severity in terms of midline crossing defects. Rescue experiments indicating that expression of specific Nova-dependent splice variants of DCC can restore midline crossing in Nova mutants, together with a number of* in vitro assays, argue that Nova and DCC function together to regulate axon growth across the midline. In the present paper, you begin by showing that Nova1/2, DCC triple mutants have even stronger defects than DCC or Nova alone, suggesting that Nova and DCC act in parallel (Figure 1).Moreover, why should both over-expressing DCC, and reducing Robo lead to a suppression of the Nova phenotypes? This is particularly puzzling given that in your prior work, you found that in contrast to expression of DCC (which rescues Nova mutants) that expression of Robo3 (which should attenuate Robo repulsion to some degree) has no effect (2016, Figure 8). In contrast, in the current work, you detect trans-heterozygous interactions between Robo3 and the Nova genes that reveal striking LOF phenotypes and suggest that Nova functions with Robo3 (Figure 2).The reviewers thus urge you to reconcile this apparent disconnect in the Discussion section, especially the new genetic interactions with Robo3, since your previous study argued for the specificity of Nova function with DCC and showed that Robo3 was unable to rescue Nova phenotypes. You should also consider including DCC in your current model in Figure 8.

This is indeed an important issue and we have revised the Results section, the Discussion section and the model in Figure 8 to better demonstrate the relationship between Nova and Dcc, and between Nova and Robo. To briefly summarize the changes here, we have found that the midline crossing defects in Nova1/2 mutants are twofold. One involves the reduction of DCC attraction, which impairs axonal outgrowth and causes fewer axons to reach the midline. The other is blocked crossing by axons that are able to reach the midline, which we show in this study results from elevated ROBO repulsion. In Dcc; Nova1/2 triple KO, both defects are manifested, in that Dcc KO blocks axons from reaching the midline (Nova1/2 dKO is loss of function but not completely null in this aspect), and elevated ROBO in Nova1/2 dKO blocks half of the axons at the midline from entering. We previously showed that restoring DCC_long_ expression in Nova mutants is able to rescue axonal outgrowth and axonal attraction toward the midline (i.e. we quantified the amount of axons that were able to arrive at the midline in 2016). Here, we further show that restoring Robo isoform expression, by reducing e6b+ levels, enables midline entry within axons that are able to reach the midline. By contrast, overexpressing Robo3.1, which represses elevated Robo repulsion, is not sufficient to restore axonal outgrowth and attraction toward the midline (2016). Instead, Robo3 and Nova1/2 have a synergistic interaction in allowing midline entry, as both groups of genes downregulate ROBO activity. We also considered the possibilities that elevated ROBO repulsion can reduce the amount of axons reaching the midline and that dampened DCC attraction can block crossing. Given that projecting toward the midline can be restored by DCC isoform expression and that blocked crossing can be largely rescued by restoring Robo isoform expression, the alternative scenarios are unlikely to make major contributions.

[Editors' note: further revisions were requested prior to acceptance, as described below.]

*The manuscript has been improved and the reviewers thought your amendments were appropriate and sufficient to move forward. They continue to be concerned that the read-outs for phenotypes from dye tracing or antibody labeling to indicate bundle thickness, etc. are semi-quantitative, and that measurements of Robo1/2 mRNA and protein levels are limited by analysis of whole spinal cord rather than relevant cell populations within the spinal cord. In addition, questions remain about the differential action of the 6b+ vs 6b- isoforms. Nonetheless, they feel that the study does highlight a critical function of a micro exon* in vivo.However, there are some remaining issues that need to be addressed before acceptance, as outlined below. As Reviewing Editor, I found that the Abstract, Discussion section and model in Figure 8 need further revision; the revisions listed would make the results more accessible and exhibit your story in the best light.1) Your summary of results in the Introduction is very clear: "Here we report that Robo1/2 alternative splicing at microexon 6b is crucial for axon guidance and is controlled by the NOVA (Neuro-oncological ventral antigen) family of splicing factors…. We show that loss of Nova1/2 alters the expression of exon 6b and leads to severe midline crossing and postcrossing guidance defects. Genetically restoring the expression profile of Robo1/2 exon 6b is able to reverse these defects in Nova mutants. Interestingly, exon 6b alternative isoforms display distinct guidance activities and their production is developmentally regulated. Consequently, ROBO-mediated repulsion is not uniformly repressed precrossing as previously believed, but is instead activated initially to block premature crossing and is sufficiently blocked during crossing."The Abstract, however, is very vague and does not inform the reader on the results. You do not indicate that microexon 6b splice variants of Robo1/2 are controlled by Nova1/2. The abstract should be rewritten to be more precise and reflective of the results.

We have rewritten the Abstract to summarize our results more precisely.

2) Likewise, the beginning of the Discussion section should summarize the results more concisely, as in the paragraph at the end of the Introduction. The remainder of the Discussion section wanders and there is a degree of hand-waving. Your findings and the advances they make in the control of DCC-Netrin and Robo-Slit signaling should be stated more clearly.

We have rewritten the beginning of the Discussion section to better summarize the significance of our findings. We have removed one paragraph from the end of the Discussion section to make the text more focused.

3) The model. In Figure 8, though very welcome, is not graphically clear; the reader should be able to glean the results without looking at the legend.a) "Mutant" should be "Mutants", as more than one mutant is included.b) 1. and 2. titles should either be put on the midline (so as to refer to both Wildtype and Mutants) or the font should be biggerc) Robo1/2 inhibition in red, and "low repulsion" are confusing. Put a space between axon under DCC and Robo1/2.d) The graphic for Microexon 6b alternative splicing being developmentally controlled is not obvious.

We have modified the Figure to address all the issues listed above.

4). Figure 8 legend is not at all clear; it attempts to compare wildtype and mutant but does not indicate wildtype vs mutant.a) For example, Figure 8 legend: "Nova1/2 regulate (should be "regulates") Dcc alternative splicing (of what?), and (should be: "but in the Nova1/2 dKO", Nova deficiency reduces Dcc_long_ and causes fewer axons to reach the midline".b) On the right of the diagram under Mutant, as drawn, it is not clear whether premature turning occurs with Robo3 KO or Nova1,2 dKO (when Robo1/2 (e6b+ increases), or whether there might be an interaction between robo3 and Robo1/2 and e6b+.

We have modified the figure legend to better compare the WT and the mutants. In the text, we use Nova1/2 to refer to both Nova1 and Nova2. We realized this is unclear to the readers, and we have clarified this point at the beginning of the Introduction.

5) Text in English. We appreciate that you have utilized an English-speaking colleague to help edit the manuscript. However,a) There are still many instances in this revision of inappropriate English usage, as in subsection “Microexon 6b splice variants of ROBO1/2 have distinct guidance activities”, Discussion section – "Consistently", should be "Consistent with these findings…" or similarFigure 5 legend – "orientated" to the left should be "oriented".b) Throughout, there is incorrect use of singular vs plural.

We have corrected the mistakes listed above. In addition, we have had Heidi Chial, a Writing Specialist from BioMed Bridge, edit our manuscript.